# Coupled modelling of subglacial hydrology and calving-front melting at Store Glacier, west Greenland

Samuel J. Cook[1], Poul Christoffersen[1], Joe Todd[2], Donald Slater[3], Nolwenn Chauché[4]

[1]Scott Polar Research Institute, University of Cambridge
[2]Department of Geography and Sustainable Development, University of St Andrews
[3]Scripps Institution of Oceanography, USA
[4]Access Arctic, Le Vieux Marigny 58160, Sauvigny les Bois, FRANCE

*Correspondence to*: Samuel J. Cook (sc690@cam.ac.uk)

**Abstract.**

We investigate the subglacial hydrology of Store Glacier in West Greenland, using the open-source, full-Stokes model Elmer/Ice in a novel 3D application that includes a distributed water sheet, as well as discrete channelised drainage, and a 1D model to simulate submarine plumes at the calving front. At first, we produce a baseline winter scenario with no surface meltwater. We then investigate the hydrological system during summer, focussing specifically on 2012 and 2017, which provide examples of high and low surface-meltwater inputs, respectively. We show that the common assumption of zero winter freshwater flux is invalid, and find channels over 1 m$^2$ in area occurring up to 5 km inland in winter, and that the production of water from friction and geothermal heat is sufficiently high to drive year-round plume activity, with ice-front melting averaging 0.15 m d$^{-1}$. When the model is forced with seasonally averaged surface melt from summer, we show a hydrological system with significant distributed sheet activity extending 65 km and 45 km inland in 2012 and 2017, respectively; while channels with a cross-sectional area higher than 1 m$^2$ form as far as 55 km and 30 km inland. Using daily values for the surface melt as forcing, we find only a weak relationship between the input of surface meltwater and the intensity of plume melting at the calving front, whereas there is a strong correlation between surface-meltwater peaks and basal water pressures. The former shows that storage of water on multiple timescales within the subglacial drainage system plays an important role in modulating subglacial discharge. The latter shows that high melt inputs can drive high basal water pressures even when the channelised network grows larger. This has implications for the future velocity and mass loss of Store Glacier, and the consequent sea-level rise, in a warming world.

## 1. INTRODUCTION

The Greenland Ice Sheet (GrIS) is currently losing mass at about 260 Gt a$^{-1}$ (Forsberg et al., 2017) and this rate has been accelerating (Kjeldsen et al., 2015). Around half of this loss is tied to ice-sheet dynamics (van den Broeke et al., 2016) and the accompanying flow acceleration is partly due to tidewater outlet glaciers, which drain 88% of the ice sheet (Rignot and

Mouginot, 2012). As such, understanding how these tidewater glaciers may change over time is crucial to our ability to predict the likely evolution of the GrIS in a warming climate.

One area of particular concern is the subglacial hydrology of these tidewater glaciers. Whilst there have been many studies focusing on the subglacial hydrology of land-terminating portions of the GrIS and its complex effect on the flow of the overlying ice (Chandler et al., 2013; de Fleurian et al., 2016; Christoffersen et al., 2018; Gagliardini and Werder, 2018; Meierbachtol et al., 2013; Sole et al., 2013; Tedstone et al., 2013, 2015; van de Wal et al., 2015), the hydrology of tidewater glaciers has received much less attention (e.g. Schild et al., 2016; Sole et al., 2011; Vallot et al., 2017), owing to the greater difficulty of gathering observations in the fast-flowing marine-terminating environment. Given the range of other processes operating at such glaciers, such as submarine melting, fjord circulation and calving, it is also much harder to disentangle and infer hydrological evolution from changes in surface velocity, though attempts have been made (Howat et al., 2010; Joughin et al., 2008; Moon et al., 2014). Direct basal observations on marine-terminating outlets were until recently limited to boreholes drilled near Swiss Camp and the lateral margin of Jakobshavn Isbræ (Lüthi et al., 2002). Only one study has, to date, reported direct observations from boreholes drilled along the central flowline, to the base of a marine-terminating glacier in Greenland. In that study, a persistently high basal water pressure of 93-95% of ice overburden indicates a largely inefficient basal water system 30 km inland from the calving margin at the fast-flowing and heavily-crevassed Store Glacier (Store) (Doyle et al., 2018). Yet, observed seasonal velocity fluctuations on the same glacier are consistent with the development of a channelised basal drainage system closer to the margin (Young et al., 2019), which calls for a physical model to spatially and temporally constrain the formation of different types of basal drainage system.

Hydrological work on marine-terminating glaciers has so far focused on the subglacial discharge that drives convective plumes in the marine terminus environment, and how this process can promote calving by undercutting the glacier through submarine melting and fjord circulation (Carroll et al., 2015; Cowton et al., 2015; Fried et al., 2015; Jackson et al., 2017; Jouvet et al., 2018; Slater et al., 2018). In particular, the state of the subglacial hydrological system is thought to be a key control on the rate and spatial distribution of submarine melting, with channelised drainage favouring the highest localised melt rates, though distributed drainage may produce the highest total volume of submarine melting, with lower melt rates that affect a larger portion of the calving front (Fried et al., 2015; Slater et al., 2015). However, our observations of the near-terminus subglacial hydrological system remain extremely limited; we can only make inferences about the location and presence of subglacial channels from the presence of plumes at the fjord surface (Schild et al., 2016), subsurface incisions into the calving front (Fried et al., 2015) and oceanographic observations (Stevens et al., 2016).

Given the paucity of direct observations, insights to marine-terminating glaciers' interaction with the ocean may be found through the integration of subglacial hydrology within physically-based models of ice flow (e.g. Banwell et al., 2013; de Fleurian et al., 2014; Hewitt et al., 2012; Hoffman et al., 2016; Werder et al., 2013). So far, these models have, however, been applied largely to land-terminating catchments in Greenland or elsewhere, where validation is easier due to the availability of better observations of the hydrological system. There is, though, no fundamental reason why they should not also function effectively in a tidewater setting. On tidewater glaciers, seasonal flow variations (Moon et al., 2014) and elevation changes

(Csatho et al., 2014) are observed too far inland to be explained purely by forcing at the glaciers' termini (Todd et al., 2018). With the advent of the Subglacial Hydrology Model Intercomparison Project (SHMIP) (de Fleurian et al., 2018), greater confidence in the results of these models is now possible, which provides further motivation to apply them in this novel manner. This would then provide the ability to dynamically model tidewater-glacier subglacial hydrology, allowing better prediction

of plume and calving activity at the front, and ice flow inland, ultimately leading to improved constraints on future sea-level rise scenarios. In this study we therefore apply a subglacial hydrological model to a large Greenland tidewater outlet glacier with the goals of A) characterising the basal drainage system, including the extent to which it may become efficient, and B) investigating how subglacial discharge drives melting at the glacier's terminus when convective plumes develop. This study therefore couples a subglacial hydrology model with a 1D plume model within the ice-flow model, Elmer/Ice (Gagliardini et

al., 2013), in order to simulate the seasonal variation in the subglacial hydrological network of Store and the resulting plume melting.

## 2. DATA AND METHODS

The study site (Sect. 2.1.), individual modelling components (Sect. 2.2.-2.4.) and their relation to each other within the model set-up (Sect. 2.5.) are described below, followed by details of the datasets used to prescribe boundary conditions (Sect. 2.6.).

This paper presents coupled subglacial hydrology and plume models within a full-Stokes 3D model of Store. The subglacial hydrology model is GlaDS (Werder et al., 2013), the ice flow model is Elmer/Ice (Gagliardini et al., 2013) and the plume model is a 1D line plume (Slater et al., 2016). Each of these is described further in turn below.

## 2.1. Study site

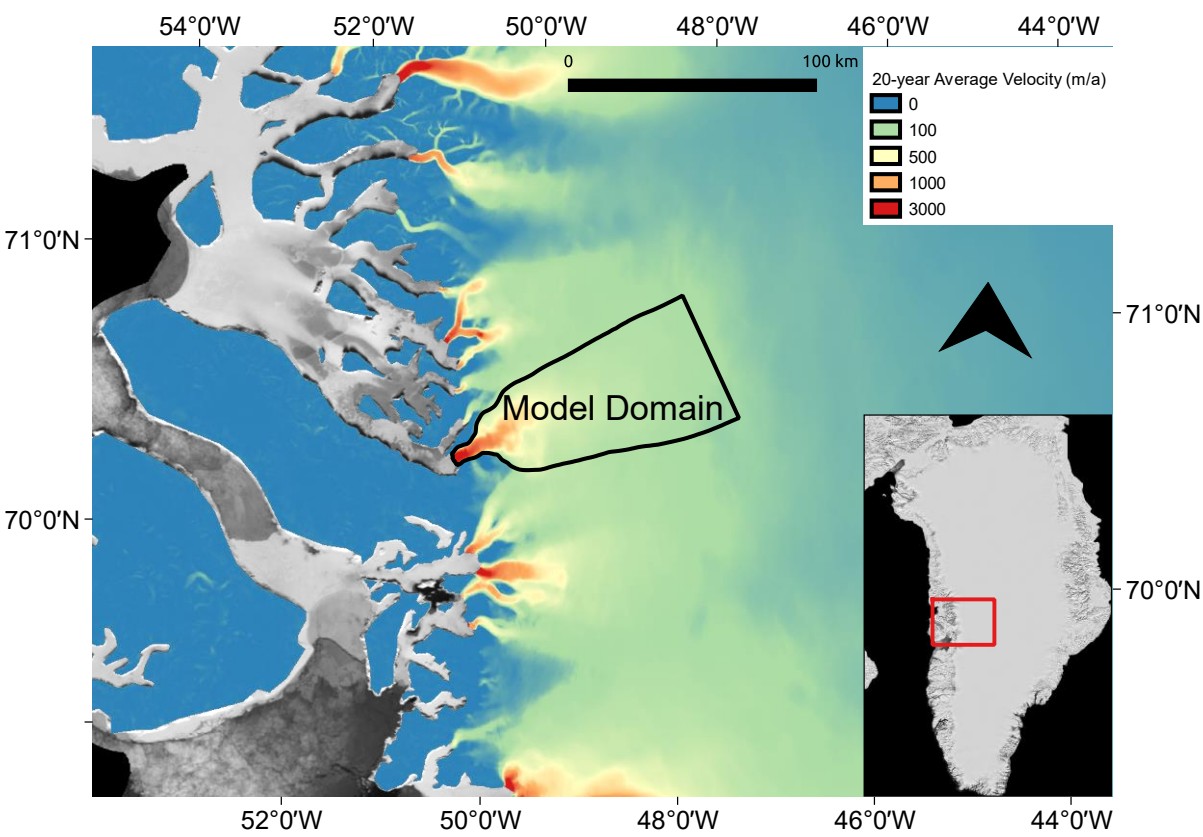

**Fig. 1 – Location of Store (inset) and model domain. Background shows the 20-year velocity average from the MEaSUREs dataset (Joughin et al. 2016, 2018).**

Store Glacier (Store), one of the largest tidewater outlet glaciers on the west coast of Greenland (70.4°N, 50.55°W), flows into Ikerasak Fjord (*Ikerasaup Sullua*) at the southern end of the Uummannaq Fjord system (Fig.1). The calving front is 5 km wide, with surface velocities reaching up to 6600 m a$^{-1}$ (Joughin, 2018), and is pinned on a sill making the terminus position relatively stable despite the trunk of the glacier flowing through a deep trough extending to nearly 1000m below sea level (Rignot et al., 2015). With no observed retreat since 1985 (Catania et al., 2018), the glacier represents a stable Greenland outlet glacier and

is an ideal target for modelling studies aiming to understand such glaciers, as the effects of rapid retreat do not need to be disentangled from 'natural' behaviour (e.g. Morlighem et al., 2016; Todd et al., 2018; Todd and Christoffersen, 2014; Xu et al., 2013).

### 2.2. Elmer/Ice ice-flow model

    The 3D, open-source, full-Stokes, finite-element model, Elmer/Ice (Gagliardini et al., 2013), is used to simulate ice flow

through solution of the Stokes equations. Elmer/Ice also provides the framework in which the other model components (below)

are implemented. For a detailed description of Elmer/Ice, readers are directed to Gagliardini et al. (2013). The model presented here also builds on work applying Elmer/Ice to tidewater glaciers presented in Todd et al. (2018). The upstream limit of the model domain was taken as the 100 m a$^{-1}$ velocity contour (Fig. 1), with a boundary condition on the inflow boundary specifying observed velocity. No flow was allowed through the lateral boundaries of the domain, which also had a no-slip

boundary condition applied, and a sea-pressure condition was specified on the fixed calving front, as well as on the basal boundary. A geothermal heat flux of 55 mW m$^{-2}$ (Martos et al., 2018) was specified at the base, and ice temperature at the upper surface (including the inflow boundary) was set equal to observations. A simple Weertman-type sliding law was applied at the base, as shown in Eq. (1):

$$\tau_b = \beta u_b$$

10                                                                                                                                                          **(1)**

Where $\tau_b$ is the basal stress, $\beta$ is the basal slip coefficient, and $u_b$ is the basal velocity.

The model mesh was refined to reach the maximum resolution of 100 m near the calving front, coarsening gradually to 2 km beyond 20 km inland. The grounding line was set to the model outflow boundary, as we do not permit the glacier to float in this study. This is not a fully realistic representation of the situation at Store, where the southern part of the terminus is floating,

but a more realistic treatment would require a substantial addition in model complexity to include the relevant calving-related processes, which are not our focus here, having been investigated by Todd et al. (2018) within a similar framework at the same glacier. We consider the impact of this simplification on our results to be negligible, with the minor exception of some aspects of the plume modelling. This impact is discussed further below.

## 2.3. GlaDS hydrology model

Modelling of Store's subglacial hydrology is achieved using the GlaDS (Glacier Drainage System) module within Elmer/Ice, an implementation of the GlaDS model (Werder et al., 2013), which participated in the SHMIP tests (de Fleurian et al., 2018) and has been developed specifically for glaciological contexts (Gagliardini and Werder, 2018; Werder et al., 2013). GlaDS simulates both a continuous sheet of water across the entire model domain, representing inefficient distributed drainage, and discrete channel elements, which can form along the edges of the mesh elements when sheet thickness locally exceeds a

threshold, thereby representing efficient drainage.

GlaDS is run on a 2D mesh distinct from the 3D ice-flow mesh, but replicating the footprint of the ice-flow mesh. That is, the nodes of the hydrology mesh are distributed across the same area as the ice-flow mesh, but at a different resolution. This allows a progressively finer GlaDS mesh resolution, starting at 100 m in the lowermost 20 km of the domain and coarsening to 2 km only in the uppermost portion of the domain, beyond 100 km from the front. Hence, we obtain a detailed understanding of the

hydrology in the main trunk of the glacier, without increasing the computational cost of calculating the velocity and temperature of the ice throughout the model domain. This dual-mesh approach requires interpolation of variables between the two meshes (the ice velocity and normal stress, along with the residual from the temperature solver and the position of the

grounding line). Channels are not allowed to form along the boundaries of the hydrology mesh and no water flow is assumed to occur across the lateral or inflow boundaries. In addition, the hydraulic potential ($\phi$) is set to 0 at the calving front (i.e. we assume the calving front is at flotation), following Eq. (2) and (3):

$$\phi = \rho_w g Z + P_w$$

**(2)**

$$P_w = \rho_w g (Z_{sl} - Z)$$

**(3)**

Where $\rho_w$ is the density of water at the calving front (i.e. of seawater in this case), $g$ is the gravitational constant, $Z$ is the elevation with respect to sea level, $P_w$ is the water pressure, and $Z_{sl}$ is sea level. In the case where $Z_{sl}$ is set at 0.0, as it is here,

and $Z$ is negative, which will be true for the outflow of the subglacial hydrological system at the bottom of the calving front, it can be seen that substituting Eq. (3) into Eq. (2) will give a result of 0 for $\phi$.

Water entering the hydrological system is derived from surface and basal meltwater production. Specifically, the source term for the hydrological model at each node on the mesh is the sum of basal and internal melting due to friction and strain, and surface melt. Basal and internal melting are computed from the interpolated temperature residual, while surface melt is taken

from a raster of melt values, as described in Sect. 2.6. below. Because the study focuses on the hydrology of the terminus region, we make the simplifying assumption that surface melt travels straight to the bed at the point of production, which is reasonable on a heavily crevassed glacier such as Store. While some runoff in reality is routed and stored at the surface (Smith et al., 2015), supraglacial stream networks are typically much smaller in size compared to the basal system considered here.

Parameters for the hydrological model are similar to those detailed in Gagliardini and Werder (2018), and are set out in Table

1. We use a higher bedrock bump height and cavity spacing, given the observed smoother sedimentary topography of Store and the length scale over which it varies (Hofstede et al., 2018). For full mathematical details of the GlaDS model and a sensitivity analysis of the model to these parameters, readers are directed to Werder et al. (2013), and for additional details on its implementation within the Elmer/Ice framework, to Gagliardini and Werder (2018). An additional sensitivity analysis was not undertaken here as being beyond the scope of this study.

The coupling between the ice-flow model and the hydrology model in this study is only one-way – there is no feedback from the hydrological system to the overlying ice – as our intention in this study was to investigate the hydrological system in winter, its expansion in summer and how its evolving nature affects submarine melting of the calving front. A coupling of ice dynamics and hydrology is beyond the scope of this study, but will be undertaken as part of future work.

**Table 1 – Parameters used in GlaDS model for all model runs in this study.**

| Description | Symbol | Value | Units |
|---|---|---|---|
| **Pressure melt coefficient** | $c_t$ | $7.5 \cdot 10^{-8}$ | K Pa$^{-1}$ |
| **Heat capacity of water** | $c_w$ | 4220 | J kg$^{-1}$ K$^{-1}$ |
| **Sheet flow exponent** | $\alpha_s$ | 3 | |

| | | | |
|---|---|---|---|
| **Sheet flow exponent** | $\beta_s$ | 2 | |
| **Channel flow exponent** | $\alpha_c$ | 5/4 | |
| **Channel flow exponent** | $\beta_c$ | 3/2 | |
| **Sheet conductivity** | $k_s$ | 0.0002 | m s kg$^{-1}$ |
| **Channel conductivity** | $k_c$ | 0.1 | m$^{3/2}$ kg$^{-1/2}$ |
| **Sheet width below channel** | $l_c$ | 20 | m |
| **Cavity spacing** | $l_r$ | 100 | m |
| **Bedrock bump height** | $h_r$ | 1 | m |
| **Englacial void ratio** | $e_v$ | 10$^{-4}$ | |

## 2.4. Plume model

For the purposes of this study, a 1D plume model based on buoyant plume theory (Jenkins, 2011; Slater et al., 2016) was implemented in Elmer/Ice. The model simulates the evolution of subglacial runoff after it emerges from the grounding line

and rises towards the fjord surface, mixing turbulently with the warm surrounding fjord water and stimulating melting at the ice-ocean interface. This model has been successfully used to model proglacial plumes in studies of diverse focus (Hopwood et al., 2018; Slater et al., 2017), including within the MITgcm ocean circulation model (Cowton et al., 2015).

In this study, a continuous sheet-style 'line' plume (Jenkins, 2011; Slater et al., 2016), split into coterminous segments, is simulated across the calving front. Our limited observational constraints currently support this line plume geometry as the most

appropriate for use at tidewater glaciers (Fried et al., 2015; Jackson et al., 2017). Discharge at each node on the grounding line is taken as the sum of channel and distributed sheet discharge within the hydrological model GlaDS. This allows the plumes and the consequent modelled submarine melt rates across the calving front to vary dynamically as the subglacial drainage system evolves over the course of each simulation, without having to specify fixed plume locations. The drag coefficient (Cd) within the plume model was increased to 0.02, following Ezhova et al. (2018). A full description of the plume model can be

found in Slater et al. (2016).

Results from the plume model are largely independent of the mesh resolution at the calving front. The input discharge and consequent submarine melt rate is calculated per metre width along the front, hence a coarser mesh will lead to more discharge at each grounding-line node. This discharge increase will, however, be spread over a larger area between nodes, so the overall input discharge and output melt rate are similar for different mesh resolutions. The frontal mesh resolution and plume segment

width used on the hydrology mesh (100 m) were chosen to fit with the frontal mesh resolution on the ice mesh to minimise interpolation artefacts and also as representing a reasonable order-of-magnitude length scale for the size of subglacial channel outlets on tidewater glaciers (Fried et al. 2015; Jackson et al., 2017). We consider this to be a reasonable simplifying

assumption, given the current lack of observational constraints for the morphology of channel outlets at the calving front of tidewater glaciers.

When discussing the results of the plume model we will make use of a number of quantities which highlight different aspects of variability between our simulations. We define the 'average melt rate' as the average over all points of the subaqueous calving front and all points in time. We define the 'mean maximum melt rate' as the average over time of the maximum melt rate at any point on the calving front. We define the 'absolute maximum melt rate' as the maximum at any time and at any point on the calving front.

## 2.5. Modelling procedure

Initially, the ice flow model was run with the simple sliding law in Eq. (1) based on an initial guess at the basal slip coefficient, $\beta$, until a converged ice temperature-velocity solution was reached. We then inverted for basal friction to match satellite-derived surface velocity at Store, producing an observationally constrained steady-state temperature-velocity solution. From this starting point, a year-long hydrological simulation was run, using only basal melt, to provide an initialised state for the hydrological system as well as the ice flow. For the subsequent hydrology runs, the timestep was set to 0.1 days and all ice dynamic variables (geometry, temperature, velocity, etc.) were kept constant, given the lack of two-way coupling and our aim in conducting this study, as discussed in Sect. 2.3 above.

Subsequent to this hydrological initialisation simulation, we performed five hydrological simulations based on different scenarios as described in Table 2. Each of these scenarios ran for three months, to replicate an actual season, and all used at least basal melt as a source term for the hydrology. One simulation (Winter) was a winter simulation, with no surface melt, meaning the simulated hydrological system carried exclusively basal melt. The remaining four scenarios describe summer simulations in which the hydrological system carries surface melt in addition to basal melt. Two of these (SummerAverage12 and SummerAverage17) used a constant surface melt equal to the average during June, July and August (JJA) in 2012 and 2017, respectively, to allow a comparison between a high-melt (2012) and a low-melt (2017) year. The final two (SummerDaily12 and SummerDaily17) instead used daily values of surface melt for 2012 and 2017 to enable exploration of the differences produced in the modelled hydrological system when using realistic transient forcing, varying day to day, compared to the summer average state. Due to the fixed geometry and ice dynamics of the hydrological simulations, the basal melt term is constant across all timesteps and all simulations, as the temperature field does not evolve, allowing easier discrimination of changes in the system caused by the addition of the surface melt in the summer simulations. The simulations are summarised in Table 2, below.

**Table 2 – Summary of hydrological simulations. BM = Basal and Internal Melt; SM = Surface Melt.**

| Name | Season | Hydrological Source | Surface-Melt Resolution |
|---|---|---|---|
| **Winter** | Winter | BM | n/a |
| **SummerAverage12** | Summer 2012 | BM + SM | Three-month average |

| | | | |
|---|---|---|---|
| **SummerAverage17** | Summer 2017 | BM + SM | Three-month average |
| **SummerDaily12** | Summer 2012 | BM + SM | Daily |
| **SummerDaily17** | Summer 2017 | BM + SM | Daily |

## 2.6. Data

The surface DEM used here to set the surface ice geometry in Elmer/Ice is from the ArcticDEM project, v2.0 (Porter et al., 2018), representing a composite view of the region between 2015 and 2018, and resampled to 25 m resolution. The basal DEM

is taken from BedMachine v3 (Morlighem et al., 2017) at 150 m resolution and a nominal date of 2007, though this was processed further to remove errors around the terminus of Store, based on work previously conducted by the authors (Todd, 2018).

Surface melt data for input to the hydrology model are runoff values from RACMO2.3p2 at 1 km spatial resolution and daily temporal resolution (Noël, 2018). Summer averages were calculated by taking the mean of the surface melt across all days in

JJA of the relevant years. The surface temperature data used as an upper-ice-surface boundary condition in the ice flow model were the average for the years 2000-2014, derived from the NASA MODIS Snow and Sea Ice Mapping Project (Hall et al., 2012, 2013). The surface velocity data used for the inversion part of the spin-up process were taken from the 20-year velocity mosaic of Greenland developed as part of the MEaSUREs project (Joughin et al., 2016, 2018). This was in order to remove the bias of any anomalous velocity patterns in any one specific year and also because sufficient good-quality velocity data for

all the years looked at in this study was not available.

The temperature and salinity of the ambient fjord water, required by the plume model, were taken from conductivity-temperature-depth casts gathered in the fjord within a few kilometres of the calving front (Chauché, 2016). Different profiles (Fig. 2) were used for summer (CTD cast from 02/08/2012; within 1 km of the calving front) and winter (CTD cast from 02/03/2013; 10 km from calving front). Both are the closest data to the calving front available, and are assumed to be

representative of conditions at the calving front.

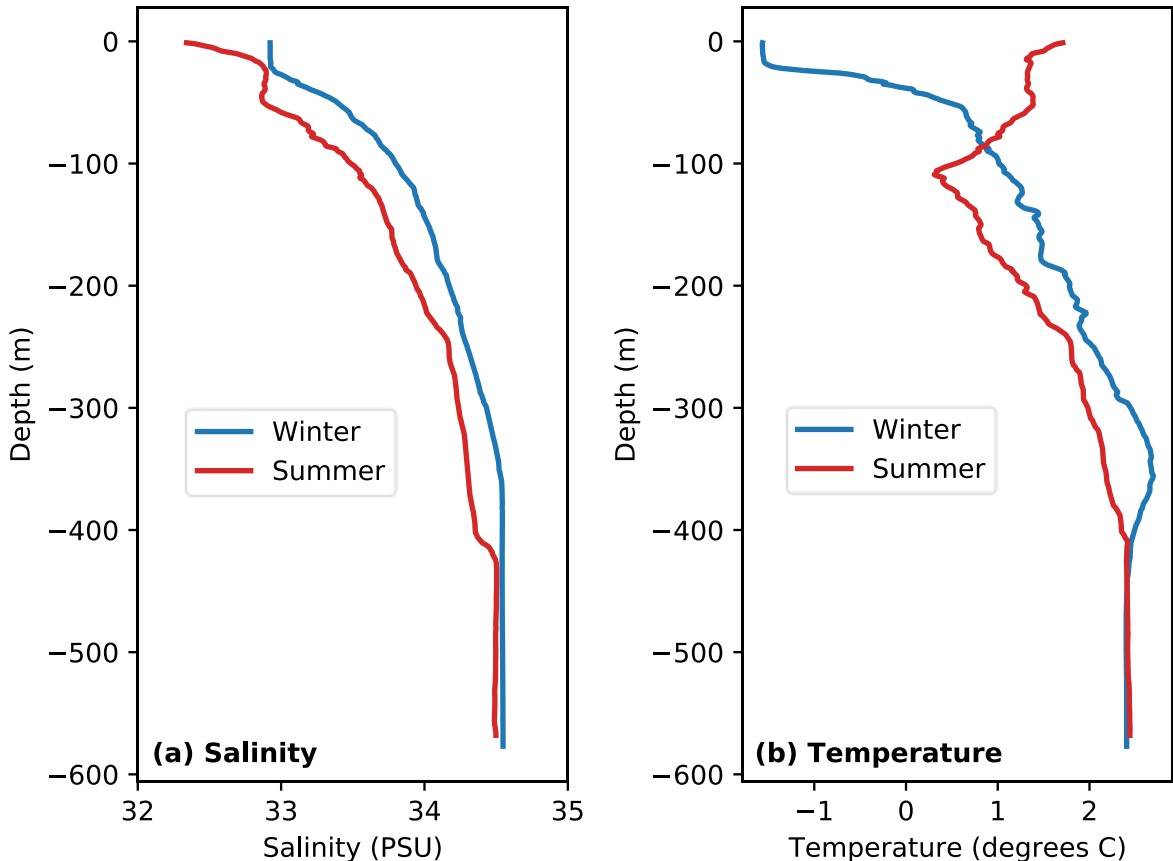

**Fig. 2 – Ambient fjord salinity and temperature profiles used as input to the plume model (Chauché 2016). Winter conditions from CTD cast on 02/03/13 approx. 10 km from the calving front; Summer conditions from CTD cast on 02/08/12 approx. 1 km from the calving front. (a) Salinity in winter and summer; (b) Temperature in winter and summer.**

## 3. RESULTS

The key simulation results are summarised in Table 3, with the important differences being picked out in the remainder of this section. The findings are then discussed in Sect. 4.

**Table 3 –  Summary of key simulation results. The channel, sheet and pressure statistics are taken from the final timestep across the entire model domain (columns marked 'End') or the timestep where maximum mean channel area in the simulation was reached (columns marked 'Max') – for the average-forced runs, the end timestep is also the max timestep, so only figures for the end timestep are shown; the plume statistics are taken from the calving front across all timesteps. 'Area Channelised' refers to the percentage of the possible channel segments occupied by channels >1 m² in area.**

|  | Winter | SummerAverage12 | SummerAverage17 | SummerDaily12 | | SummerDaily17 | |
|---|---|---|---|---|---|---|---|
|  | End | End | End | Max | End | Max | End |

| | | | | | | | |
|---|---|---|---|---|---|---|---|
| **Mean channel area (m²)** | 0.04 | 9.84 | 6.45 | 12.10 | 8.18 | 7.00 | 5.25 |
| **Mean channel flux (m³ s⁻¹)** | $8 \times 10^{-4}$ | 5.32 | 2.40 | 7.47 | 2.44 | 3.26 | 2.46 |
| **Area channelised (%)** | 0.05 | 12.05 | 6.75 | 15.26 | 10.77 | 8.40 | 6.81 |
| **Mean sheet discharge (m³ s⁻¹)** | $3 \times 10^{-4}$ | 0.090 | 0.016 | 0.104 | 0.034 | 0.023 | 0.008 |
| **Mean sheet thickness (m)** | 0.18 | 0.47 | 0.32 | 0.51 | 0.36 | 0.34 | 0.28 |
| **Mean effective pressure (MPa)** | 2.01 | 1.13 | 1.30 | 1.27 | 1.43 | 1.37 | 1.16 |
| **Mean plume melt rate (m d⁻¹)** | 0.15 | 0.68 | 0.50 | 0.65 | 0.65 | 0.50 | 0.50 |
| **Mean maximum plume melt rate (m d⁻¹)** | 0.43 | 4.25 | 3.13 | 3.65 | 3.65 | 3.01 | 3.01 |
| **Total plume melt (m³ a⁻¹ x10¹⁰)** | 1.26 | 5.85 | 4.29 | 5.95 | 5.95 | 4.36 | 4.36 |

### 3.1. Winter baseline simulation

The results from the Winter run show a varied subglacial drainage system at Store, where channels may form even in winter when the system is fed exclusively by basal melt produced by frictional and geothermal heat (Fig. 3). Channels of 1 m² or more in cross-sectional area, a threshold we found functions effectively as a discriminator for regions of significant channel growth, are found in a few regions extending up to 5 km inland from the terminus at the end of the model run. Smaller channels link these up and form the majority of an arborescent network with three main branches, reaching to 40 km inland (Fig. 3b). Channels this small are indicative of a distributed drainage system, rather than a true channelised system. We show them in Fig 3. because they illustrate where a connected subglacial drainage system will subsequently develop. One branch of the subglacial drainage system drains the northern side of the model domain, one the southern side, and one the centre. These branches then converge into one major, central flow path that splits in two near the terminus, with one flow path exiting at the northern margin of the ice front and one at the southern margin. The pattern of discernible discharge within the distributed sheet (down to 0.0001 m² s⁻¹) (Fig. 3c) is similar, which is to be expected, as the thickness of the distributed sheet (typically

up to 1 m near the terminus, progressively dropping to below 0.1 m beyond 100 km inland) determines the location of the channels within the hydrology model.

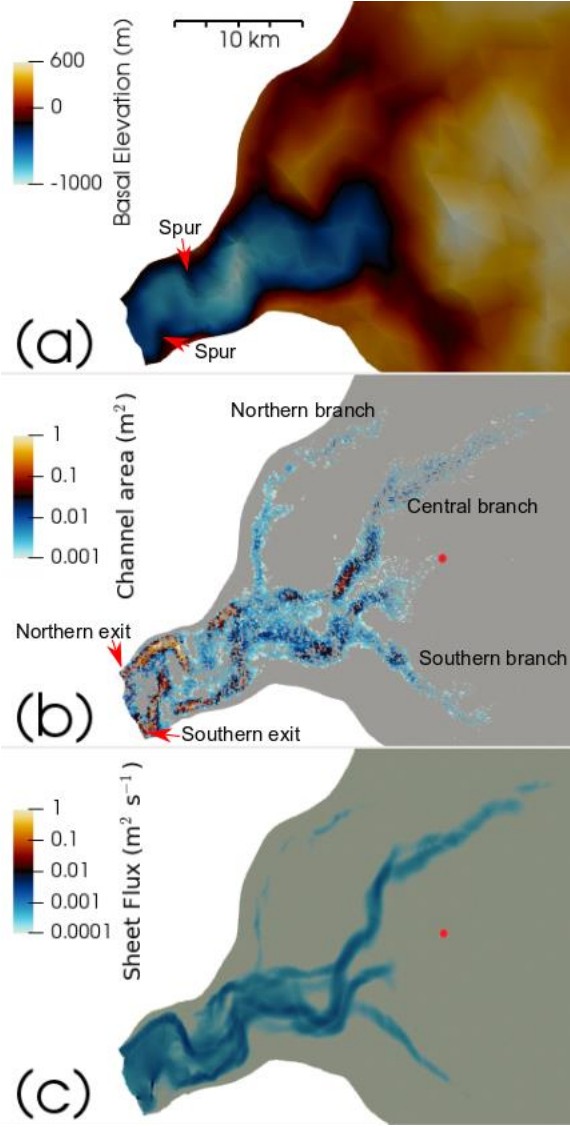

**Fig. 3 – Winter hydrological system at Store at end of Winter run and basal topography. (a) bed elevation; (b) channel area ; (c) sheet flux. The red dot in panels (a) and (b) marks the location of the S30 site from Young et al. (2019). Channels and sheet drainage pathways can be seen to largely follow the deeper parts of the bed. We have shown very small channels and low sheet discharges to fully display the existence of a connected winter drainage system.**

The discharge patterns are controlled by the hydraulic potential gradient, which, as can be seen from Fig. 3a, is mainly determined by the basal topography of Store and the ice thickness, with the farther-inland areas of greater hydrological activity following the deeper parts of the bed. The same can be seen nearer the terminus where the successive southward and northward bends in the drainage pathways upstream of the terminus are related to spurs of shallower bedrock jutting into the central trough from the northern and southern margins, respectively. These push the flow pathways towards the edges of the trough, compared to the more central flow paths farther inland.

In total, the input to the hydrological system from basal melt in winter amounts to $4.7 \times 10^7$ m$^3$ over the 92 days of the Winter simulation, or 5.96 m$^3$ s$^{-1}$, with the resulting subglacial discharge across the grounding line split 2:1 between the channels and the distributed sheet, respectively. This is sufficient to drive convective plumes and a diffuse pattern of plume-induced calving-front melting throughout the winter (Fig. 4a), with a persistent diffuse plume at depth, mainly driven by discharge from the distributed sheet with occasional enhancement from channel outlets, across most of the calving front. The absolute maximum melt rate of 1.1 m d$^{-1}$ is found at the deepest point of the calving front, where observations show that the ice becomes buoyant and floats, as shown by a marked surface depression behind the calving front denoting the flexion zoneDespite the absence of surface input, melt rates of 0.2-0.3 m d$^{-1}$ are widespread. Overall, this leads to an average plume melt rate of 0.15 m d$^{-1}$, excluding the subaerial portion of the calving front, with 0.51 m d$^{-1}$ the mean maximum melt rate. The resulting meltwater flux to the fjord from plume melting is 3.48 m$^3$ s$^{-1}$.

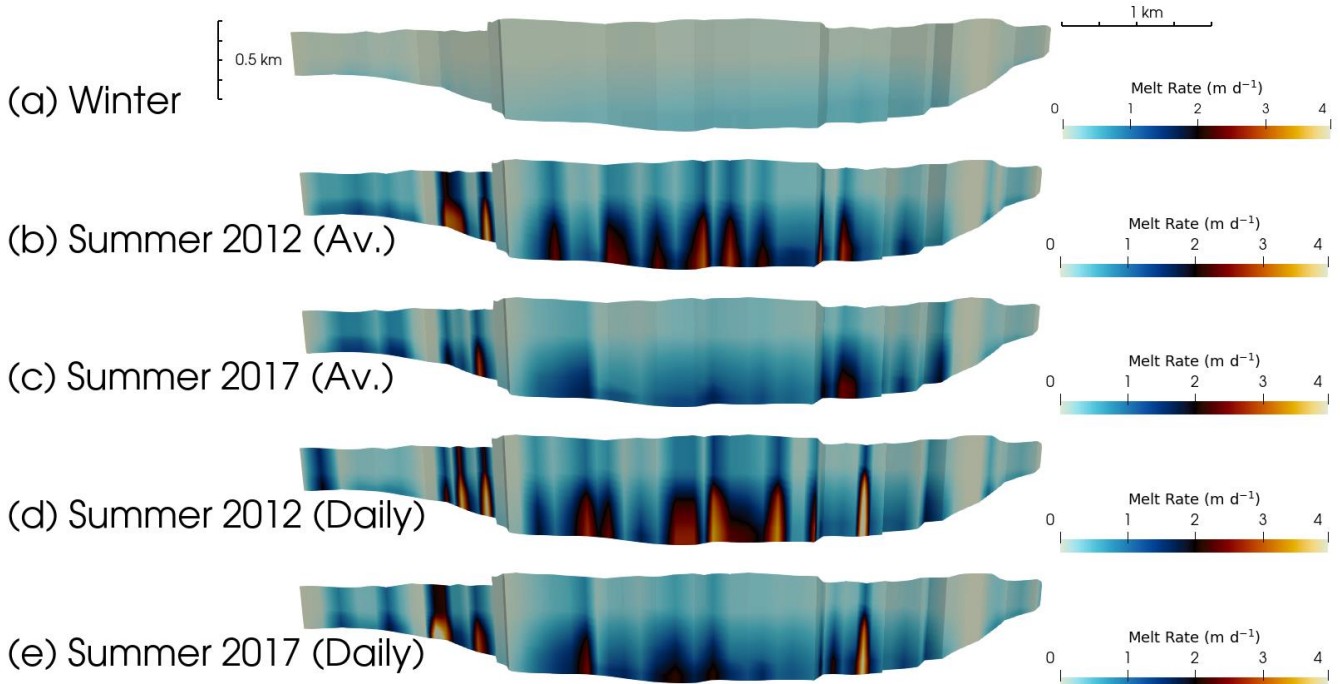

**Fig. 4 – Patterns of typical plume-generated frontal melt across all simulations, showing the 9th August for panels (b)-(e). (a) Winter run; (b) SummerAverage12 (average-forced) run; (c) SummerAverage17 (average-forced) run; (d) SummerDaily12 (daily-forced) run; (e) Run SummerDaily17 (daily-forced) run. Note how higher summer plume activity is concentrated into a relatively small number of localised high-melt plumes.**

### 3.2. Average-forced summer hydrology and plumes

In the first set of summer simulations we forced the model with RACMO surface runoff for Store averaged over JJA in 2012 and 2017. With the addition of surface meltwater, the subglacial hydrological network is found to expand substantially (Fig. 5 and 6), but does not reach a steady state by the end of either simulation. In 2012 (run SummerAverage12), the number of channel elements >1 m$^2$ in cross-sectional area grows by three orders of magnitude through the summer. Mean channel area, meanwhile, rises by two orders of magnitude, whilst mean channel flux across all sizes of channel jumps by four orders of magnitude. In 2017, when surface melt was 63% lower than in 2012, the expansion is reduced: the number of channels >1 m$^2$ in area only increases by two orders of magnitude compared to winter, with mean channel area up by two orders of magnitude again, but only reaching 6.45 m$^2$, and mean channel flux increasing once more by four orders of magnitude, but only to 2.40 m$^3$ s$^{-1}$. As the basal hydrological system accommodates surface meltwater, the channels grow significantly in size and channels over 1 m$^2$ in cross-sectional area reach farther inland – up to 55 km in 2012 (Fig. 5a), although less (30 km) in 2017 (Fig. 5b). Discernible distributed sheet discharge pathways, meanwhile, extend up to 65 km in 2012 (Fig. 6a), and growth in these is similarly reduced in 2017, to 45 km (Fig. 6b). This is reflected in the mean distributed sheet discharge figures at the end of the SummerAverage12 and SummerAverage17 model runs, which show a 226-fold and 40-fold increase on the Winter run, respectively. At the same time, the mean distributed sheet thickness increases by 259% in 2012, and by 173% in 2017,

compared to winter. The combined effect of these changes in the hydrological system is to reduce mean effective pressure (defined as ice overburden pressure minus water pressure) across the model domain by 44% in summer 2012 and by 35% in 2017; i.e. water pressures are higher in summer compared to winter.

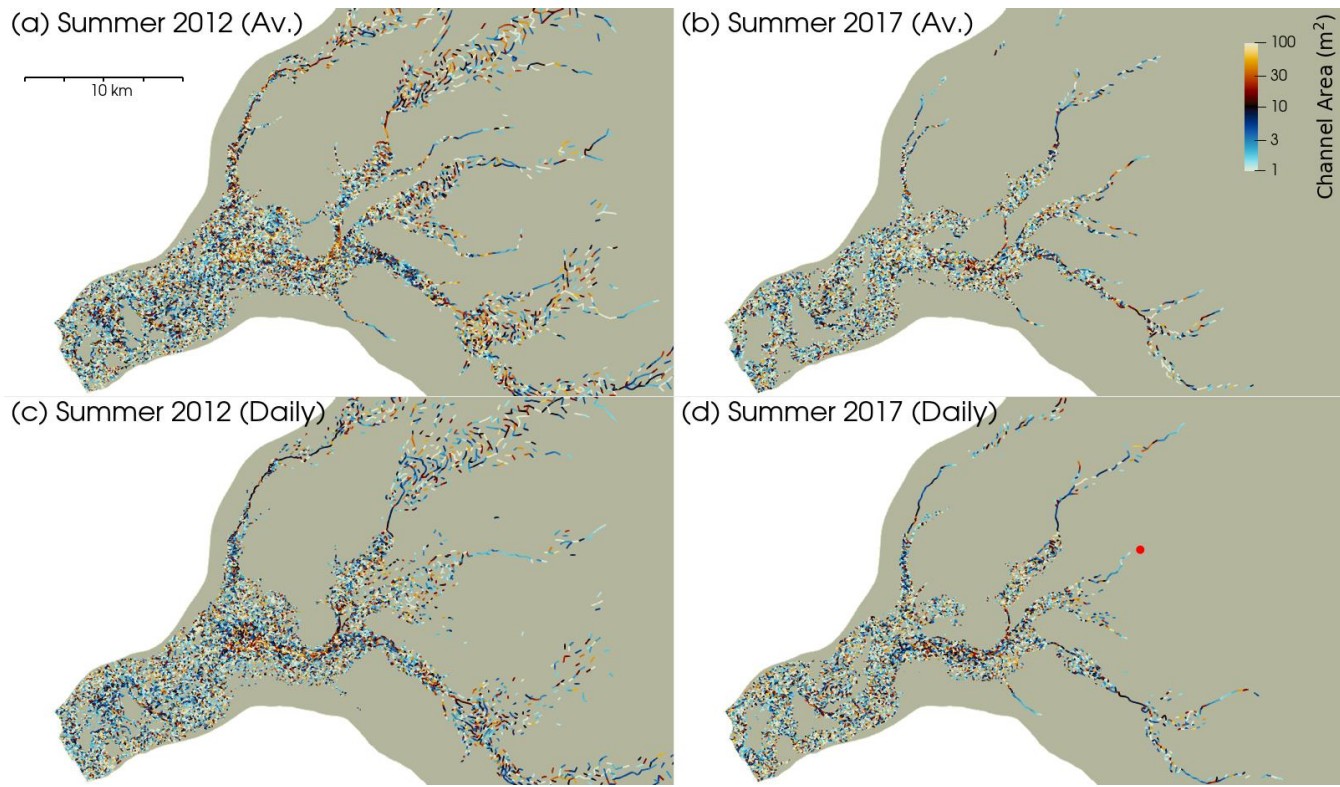

5 **Fig. 5 – Summer channel network of Store. (a) SummerAverage12 model run; (b) SummerAverage17; (c) SummerDaily12; (d) SummerDaily17 (red dot shows S30 study site from Young et al. (2019)). All the panels show the channel network at the end of the respective simulations, after three months of surface melting. The daily-forced runs show a less extensive channel network owing to declining surface melt towards the end of the melt season.**

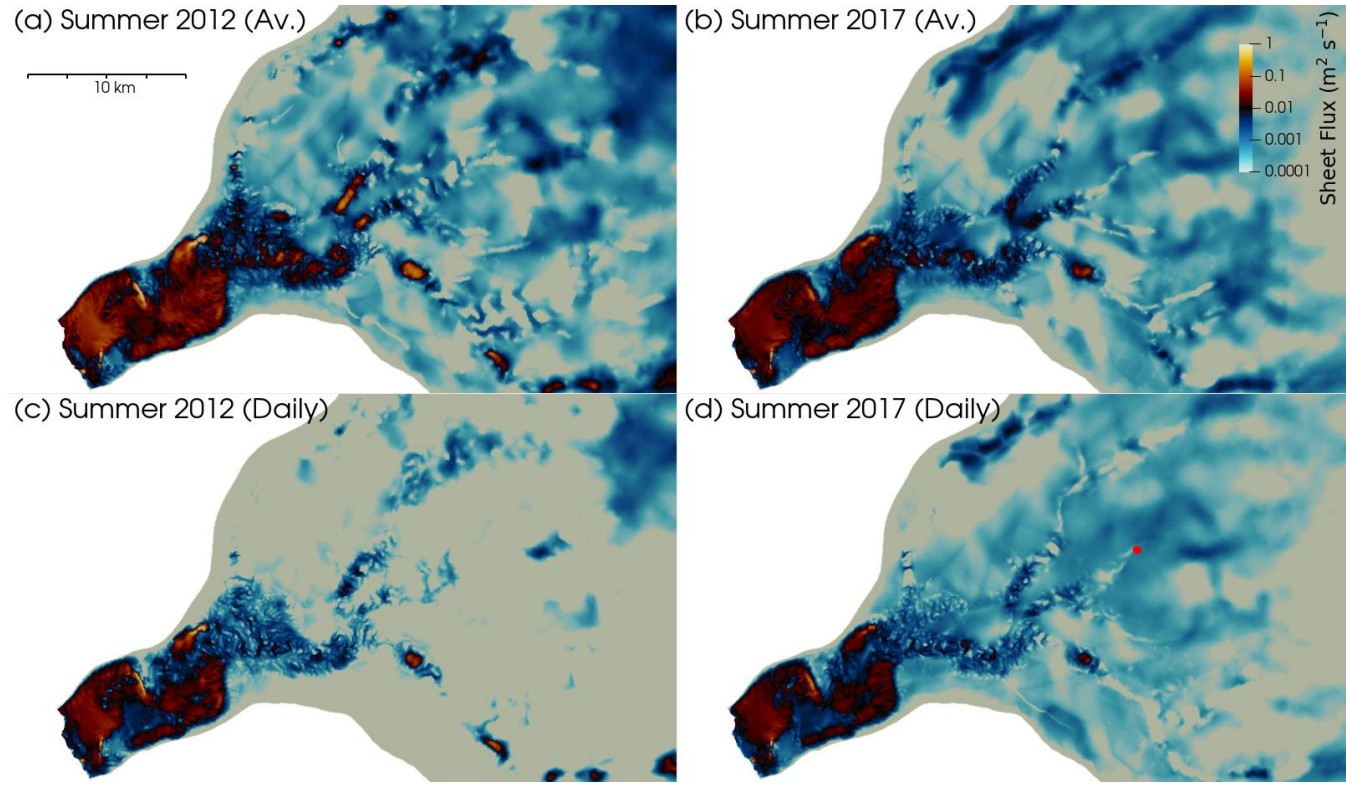

**Fig. 6 – Summer distributed sheet layer at Store. (a) SummerAverage12 model run; (b) SummerAverage17; (c) SummerDaily12; (d) SummerDaily17 (red dot shows S30 study site from Young et al. (2019)). All the panels show the sheet layer at the end of the respective simulations, after three months of surface melting. The daily-forced runs show a less extensive sheet owing to declining surface melt towards the end of the melt season.**

Plume structure and the resulting submarine melting also differ markedly between seasons and years. In summer 2012, we find strong, channel-fed plumes that usually reach the surface spaced along the majority of the calving front, with the exception of the southern extremity (Fig. 4b – the right-hand side of the terminus). In summer 2017, these stronger plumes, though still mostly reaching the surface, are more spatially restricted, appearing primarily in two regions: one on the northern side of the terminus and one around the deepest part of the calving front, where the highest melt rates are observed in winter (Fig. 4c). The resulting average melt rate for 2012 is 0.68 m d$^{-1}$, and for 2017 it is 0.50 m d$^{-1}$, rising to 4.25 and 3.13 m d$^{-1}$, respectively, for the mean maximum melt rate. Defining long-term average melt rates for areas specifically inside strong plumes or outside of them is difficult, as the location of strong convection-driven summer plumes varies as points of discharge from the hydrological system evolve, but rates of <1 m d$^{-1}$ for the diffuse, distributed-sheet-driven plume, and 2-4 m d$^{-1}$ for the stronger channel-driven plumes are typical. Absolute maximum melt rates, meanwhile, reach up to 12.6 m d$^{-1}$ for both SummerAverage12 and SummerAverage17.

### 3.3. Daily-forced summer hydrology

In the SummerDaily12 and SummerDaily17 runs we forced the model with daily values of RACMO surface runoff for Store during JJA in 2012 and 2017, respectively. Given the temporally varying nature of the surface-melt forcing, we will consider two sets of results for these runs: the end state of the simulation and the state at the maximum extent of the hydrological system. We define the latter as the time when mean channel area reaches its maximum value. Fig. 5c, d and Fig. 6c, d show the end states of the SummerDaily12 and SummerDaily17 runs as an illustration of how the different surface-melt forcings can lead to substantially different outcomes.

For the end state of SummerDaily12, the mean channel area drops by 17% compared to the SummerAverage12 run, and the mean channel flux drops by 54%, whilst the number of channel segments >1 m$^2$ in cross-sectional area drops by 11%. For the maximum state of SummerDaily12, though, the mean channel area increases by 23% compared to SummerAverage12, mean channel flux by 41%, and the channelised area by 27%. Concomitantly, the mean distributed sheet discharge and thickness at the end of the SummerDaily12 run are 63% and 25% down, respectively, compared to the SummerAverage12 run, but are 15% and 7% higher than SummerAverage12 when considering the maximum state. Overall, therefore, the change in surface-melt forcing to realistic daily totals, rather than a constant average, leads to a larger maximum drainage system extent that starts to exhibit significant decay towards the end of the melt season.

For SummerDaily17, a similar pattern is observed. At the end state of SummerDaily17, mean channel area, distributed sheet discharge and distributed sheet thickness are, respectively, 19%, 51% and 12% lower than at the end of the SummerAverage17 model run. Channel flux, along with the channelised area, show small increases of 3% and 1%, respectively, however. Considering the maximum state of the SummerDaily17 run, though, mean channel area increases by 9%, channel flux by 36%, and the area covered by channels >1 m$^2$ in cross-sectional area by 25%. Distributed sheet discharge and thickness similarly increase by 42% and 9%, respectively. The numerical values from which all these percentages are derived are given in Table 3, above. Similarly to 2012, therefore, we find the change in surface-melt forcing to produce a larger drainage system that then begins to decay as surface melt tails off.

Looking at the plume results for SummerDaily12 and SummerDaily17 (Fig. 4d, e), both daily-forced runs show a very small decline in plume activity compared to the average-forced (SummerAverage12 and SummerAverage17) runs. In the SummerDaily12 run, the average melt rate decreases by 5% on the average melt rate for the SummerAverage12 run, and the mean maximum melt rate drops by 14%. For SummerDaily17, the average melt rate only differs by 0.1% compared to SummerAverage17, but the mean maximum melt rate drops by 4%. The overall pattern of plume activity is broadly similar to that seen in the average-forced simulations, as can be appreciated by comparing Fig. 4d and e to panels b and c, with some shift in plume locations as the different forcing leads to variations in the resulting channel networks. Absolute maximum melt rates also follow suit and decrease for both simulations, reaching 9.0 m d$^{-1}$ for SummerDaily12, and 10.1 m d$^{-1}$ for SummerDaily17. The total amount of melt generated by plumes, however, increases slightly in the daily-forced simulations compared to the average-forced ones, by a little under 2% in both 2012 and 2017.

The daily-forced simulations also allow us to examine the contributions to total melt by component over time, though the basal melt, as explained in Sect. 2.5, remains constant throughout the simulations. Surface melt in SummerDaily12 is, as would be expected, the dominant factor, being one to two orders of magnitude larger than any other source of melt during the summer (Fig. 7). Plume melt, meanwhile, remains an order of magnitude greater than basal melt throughout the SummerDaily12 simulation, except for the first 20 days of the model run. Compared to the average-forced SummerAverage12 run, the more variable surface-melt input in the SummerDaily12 run also leads to greater variability in the plume melt rate. It is notable, however, that between day 72 of the model run and the end, the SummerDaily12 plume melt rate is generally equal to the SummerAverage12 plume melt rate, despite surface-melt input being somewhat lower on occasion. Similarly, the large drop in surface melt on day 72 of the SummerDaily12 run does not show any impact on plume melt at the time or afterwards. The reasons for this will be considered further in Sect. 4, below. Sheet discharge, as a proxy for the development of the subglacial hydrological system shows a sensitive, slightly lagged response to variations in surface melt in the first half of the model run, but a much more damped response in the second half. The reasons for this will also be discussed in Sect. 4.

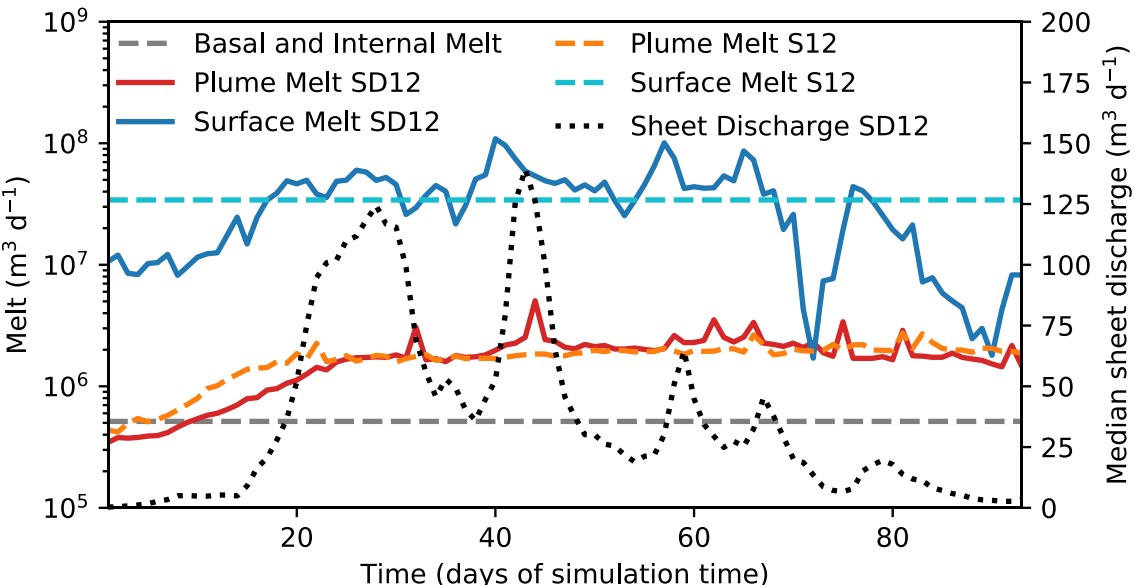

**Fig. 7 – Time series of melt sources in SummerDaily12 (red and blue solid lines) and SummerAverage12 (orange and light blue dashed lines) model runs. Note logarithmic y-axis. Basal and internal melt was constant across both runs and is included for comparative purposes – note how plume melt is of equal or greater importance. Median sheet discharge (dotted line) shows response of subglacial hydrological system to surface melt, and evolution of the system towards greater channelisation over melt season.**

For SummerDaily17 (Fig. 8), the overall pattern is similar, but the dominance of surface melt in a cooler year is reduced, with surface melt dropping below plume melt on at least two separate occasions, and even below basal melt at one point (Day 31, equivalent to the 1st July). Plume melt still exceeds basal melt throughout, except for the first 8 days (Fig. 8), underlining the importance of this mechanism even in cooler years. Similarly to summer 2012, it is also clear that, despite some periods of

low surface melt in the SummerDaily17 run, the resulting plume melt rates are comparable to those from the constant-average-forced SummerAverage17 run. Again, even with a constant forcing in the SummerAverage17 run, variable plume melting is seen, further underlining how important the underlying structure of the subglacial drainage system is in determining the resulting outflow. Unlike in 2012, however, sheet discharge remains sensitive to surface melt variations until around day 70 of the model run, exhibiting a more damped response thereafter.

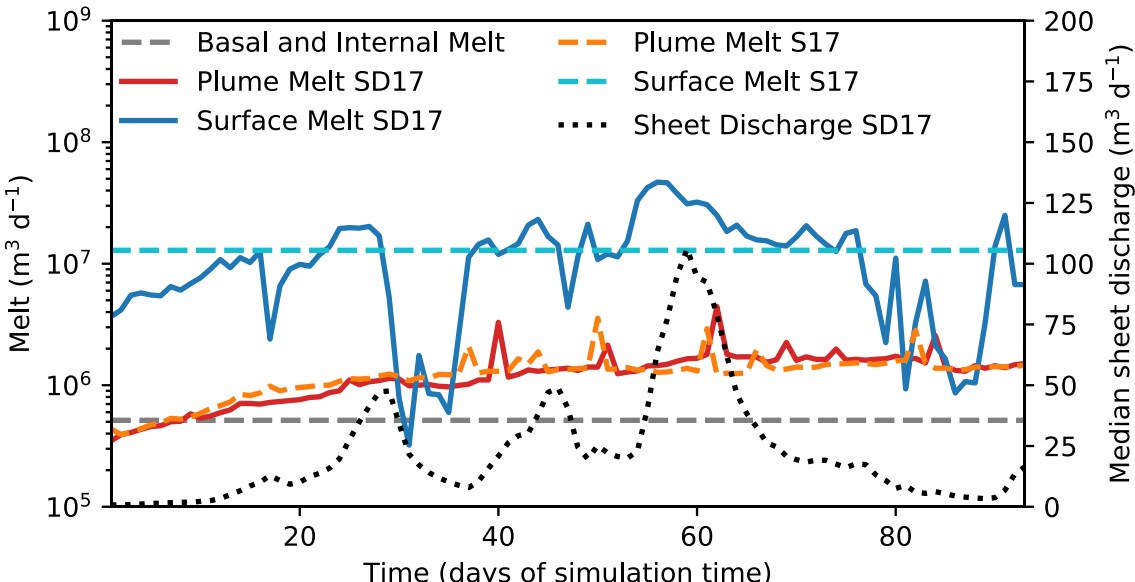

**Fig. 8 – Time series of melt sources in SummerDaily17 (red and blue solid lines) and SummerAverage17 (orange and light blue dashed lines) model runs. Note logarithmic y-axis. Basal and internal melt was constant across both runs and is included for comparative purposes – note how plume melt is of equal or greater importance. Median sheet discharge (dotted line) shows response of subglacial hydrological system to surface melt, and evolution of the system towards greater channelisation over melt season.**

## 4. DISCUSSION

### 4.1. Winter subglacial hydrology and plume activity

This study is amongst the first to constrain the nature of evolving hydrological systems beneath fast-flowing tidewater glaciers in Greenland. When the GlaDS model and Elmer/Ice are applied to Store, we predict an active subglacial drainage system consisting of channels and a distributed sheet layer to be present even in winter, when channels exceeding 1 m$^2$ in cross-sectional area form up to 5 km inland from the calving front, while a distributed sheet and smaller channels extend a further 40 km inland. This is the first time to our knowledge that the existence of such a system in winter has been shown in a model and has important implications for our understanding of tidewater glacier dynamics and the ice sheet's interaction with the ocean. In contrast to previous work, which assumed zero freshwater flux into fjords outside the summer melt season and therefore for the largest part of the year (e.g. Carroll et al., 2015; Slater et al., 2018), we demonstrate that the freshwater flux

within a channelised basal drainage system is in fact sufficient to drive convective plumes across the calving front, leading to localised melt rates of up to 1.1 m d$^{-1}$ in winter at the deepest portion of the calving front, where the strongest distributed-sheet-driven plume is modelled. Averaged across the entire subaqueous portion of the calving front, this melting equates to 0.15 m d$^{-1}$. This is below the 1.9±0.5 m d$^{-1}$ estimated by Chauché (2016) using CTD and ADCP data gathered in winter 2012-13 as inputs to the Gade (Gade, 1979) and Motyka (Motyka et al., 2003) models of fjord circulation and melting. It should also be noted that modelled melt rates from plumes consistently underestimate observed melt rates (e.g. Sutherland et al., 2019); this is a pervasive problem in plume modelling, so it is to be expected that we find a similar result. We also model an average winter subglacial discharge of only 5.96 m$^3$ s$^{-1}$, 69% of which derives from channels and 31% from the distributed sheet. This is, as expected, at the lower range of estimates (1-72 m$^3$ s$^{-1}$) presented in Chauché (2016), so our lower melt rates are consistent with this and also with the low melt rate of 0.4±0.1 m d$^{-1}$ calculated for runoff-free simulations at Store by Xu et al. (2013). Freshwater flux into the fjord from submarine melting is 3.48 m$^3$ s$^{-1}$ on average, which combined with the subglacial discharge of 5.96 m$^3$ s$^{-1}$, gives a total winter meltwater flux to the fjord of 9.44 m$^3$ s$^{-1}$. This freshwater flux may well be sufficient to drive winter-time buoyancy-driven fjord circulation, pulling warm Atlantic water towards the calving front at depth and resulting in further melting (Christoffersen et al., 2011; Mortensen et al., 2018; Straneo et al., 2010). This may be further enhanced by wind-driven circulation in autumn and early winter, when fjords are ice free and winds are strong (Christoffersen et al., 2011). Overall, though, our results for winter at Store here suggest basal meltwater production is lower and drives less intense melting at the calving front than estimated by Chauché (2016), with higher melt rates being confined to the deeper section of the calving front. This indicates either that (i) our model may lack a process that releases additional subglacial meltwater in winter, e.g. if some of the runoff from the previous melt season went into subglacial storage before it was released (Chu et al., 2016), or (ii) that we correctly predict the release of subglacial meltwater but underestimate the resulting submarine melting, perhaps due to uncertainty in the melt rate parameterisation or due to not taking account of fjord-scale circulation (Slater et al., 2018).

Our winter run results also demonstrate the critical nature of the depth of subglacial discharge for driving plume melting. In the model results, the area of highest subglacial discharge in winter is actually towards the northern margin of the calving front (left-hand side of Fig. 4), but very little plume melting is produced there. Instead, the higher melt rates are concentrated across the deepest parts of the front, where subglacial discharge is lower. This disparity can be related to the vertical profile of winter water in the fjord (Fig. 2a). For water input above a depth of 300 m, which is the case for the northern margin of the calving front, the surrounding ambient water is cold and highly stratified, so that the plume quickly reaches neutral buoyancy and what ambient water it does bring into contact with the ice front has little melting potential. For the water discharged across the south-central part of the calving front, where the depth exceeds 500 m, though, the plume is mixing with warmer, less stratified water that allows it to generate significantly more melt, which will be further enhanced by the increase in thermal energy that comes with a reduced pressure-melting point. Ambient conditions are also rather constant from the grounding line up to around 350 m, where the mid-water-column thermal maximum is reached, making it easier for the plumes to rise until they hit this lower-density layer. Therefore, if warmer water is present at depths in fjords throughout the winter, we find it likely that significant

melting occurs at depth in winter at tidewater glaciers in Greenland, even with limited subglacial discharge compared to summer.

## 4.2. Summer subglacial hydrology

When surface melting is incorporated in simulations of the summer melt season, the extent of both concentrated channels and distributed sheet systems grows substantially compared to winter (Fig. 5 and Fig. 6). Using the seasonally averaged mean surface melt between 1st June and 31st August in 2012 (run SummerAverage12), we find channels of over 1 m$^2$ in area extending to 55 km inland from the terminus (Fig. 5a), with an active distributed sheet layer again extending a further 10 km (Fig. 6a), and a resulting average freshwater flux to the fjord of 421 m$^3$ s$^{-1}$. Of the latter, 95% comes from channel outflow and 5% from plume melting, while discharge from the distributed sheet is negligible. This shows how surface melt expanded the subglacial drainage system during the warmest summer at Store in the observational record. When the model is forced by mean surface melt for the same period in 2017 (run SummerAverage17), when surface melt was much lower (149 m$^3$ s$^{-1}$ compared to 395 m$^3$ s$^{-1}$ in SummerAverage12), close to the mean for 1981-2010, we find channels of over 1 m$^2$ in area reaching 30 km inland (Fig. 5b) and the distributed sheet 45 km (Fig. 6b). Whilst the average freshwater flux to the fjord drops by 59.7% to 170 m$^3$ s$^{-1}$, the relative contributions from channels (91%), plume melting (9%) and the distributed sheet (<1%) remain largely unchanged.

In this study we also examined how the basal water in our model responded when day-to-day differences in surface melt were introduced. In 2012 (run SummerDaily12), we find the daily incorporation of surface melt to produce a larger subglacial drainage system at the system's maximum extent (Fig. 9a), with 27% more channels that are 23% larger on average and contain 41% more water on average than the end state of the system when we forced the model with seasonally averaged surface melt (SummerAverage12) (Fig. 5a). By contrast, by the end of the SummerDaily12 simulation (Fig. 5c), we find 11% fewer channels that are 17% smaller and hold 54% less water, on average, compared to the end state of SummerAverage12 (Fig. 5a). For summer 2017 (run SummerDaily17), we observe a similar pattern of a larger maximum extent (Fig. 9b) and smaller final extent (Fig. 5d) of the hydrological system compared to that seen at the end of the average-forced 2017 run (SummerAverage17) (Fig. 5b), with the exception of the number of channels and mean channel flux, which both show small increases over the final SummerAverage17 values even at the end of the SummerDaily17 run. These results for 2017 also agree well with the observations of a high-pressure distributed drainage system 30 km inland in 2014-15 reported in Doyle et al. (2018) and Young et al. (2019), towards the centre of the model domain (see Fig. 3). Young et al. (2019) posited the existence of a channelised drainage system forming up to, but not beyond this point, based on observed velocity patterns from radar and GPS measurements, with a pronounced slowdown occurring at lower elevations on Store in the summer. Doyle et al. (2018), meanwhile, suggested that persistent high pressure and rapid drainage in boreholes at the site were best explained by them tapping in to an extensive distributed drainage system. Our results for summer 2017, a better comparison for observed melt in 2014-15, concur with this pattern, with significant channel growth ceasing around the 30 km mark in the region of the study site, but with a major distributed sheet drainage pathway predicted to lie in its vicinity (red circle on Fig. 5d and 6d).

The differences in the daily-forced runs can be linked to the variability in forcing – in the SummerDaily12 run, the last two weeks of model time have steadily decreasing surface-melt forcing, with a small up-tick for the last 3 days of the run (Fig. 7). Compared to the average-forced run (SummerAverage12), it is therefore to be expected that a smaller hydrological system is found at the end of the run. A similar process is observed for summer 2017, with the drainage system in SummerDaily17

decaying as surface melt tapers off from Day 80 (equivalent to the 19th August) onwards (Fig. 8). However, unlike in SummerDaily12, there are several major surface-melt spikes after this point in the SummerDaily17 run (compare the right-hand sides of Fig. 7 and Fig. 8), explaining why channel flux and channelised area at the end of the run do not show a drop compared to SummerAverage17. Channels start to decay, as the smaller mean channel area testifies (Table 3), but the extra surface melt keeps the system from closing down as swiftly as in summer 2012. The same idea explains the finding of lower

effective pressures (and therefore higher water pressures) at the end of the SummerDaily17 run, compared to the end of the SummerDaily12 run (Table 3), despite the lower melt input in 2017. The surface-melt spikes in the last two weeks of the SummerDaily17 run (Fig. 8) re-pressurise the decaying system, whereas the smoother tapering off in SummerDaily12 (Fig. 7) means the decaying drainage system remains at over-capacity and keeps water pressures lower. This interpretation is reinforced by the evolution of sheet discharge in the two summers. In 2012, the strong response to surface melt variations in the first half

of the model run shows a predominantly distributed hydrological system with most water transiting through the sheet; the more damped response in the second half shows the formation of a predominantly channelised system where water is preferentially routed through the efficient channels rather than the inefficient sheet. The lagged nature of the sheet's response, however, means it is not possible to see how it responds to the increased melt at the very end of the SummerDaily12 run. In 2017, the pattern of sheet drainage response shows widespread channelisation was not established until towards day 70 (9th August), but

was maintained until the end of the model run, as there is little response of sheet drainage to the surface melt fluctuations from day 80 (19th August) onwards.

Looking at the maximum extent of the hydrological system in the daily-forced runs (Fig. 9, Table 3), it is important to note the dynamism this reveals in the drainage system of Store. For SummerDaily12, maximum extent is reached on Day 60 (equivalent to July 30th) of the model run, with little growth after day 45 (15th July); for SummerDaily17, on Day 75 (equivalent

to August 14th), levelling off from day 63 (2nd August), according with the onset of widespread channelisation shown by the sheet discharge time series (Fig. 7, Fig. 8) as described above. Within a month, these systems, which are substantially larger than those achieved by the end of the average-forced SummerAverage12 and SummerAverage17 runs (Table 3), die back considerably as melt inputs drop. Of particular interest is also the timing of maximum system extent versus that of maximum melt input. For SummerDaily12 (Fig. 7), the maximum melt input is achieved on Day 40 (equivalent to 10th July) of the run,

with another very similar peak at Day 57 (27th July). For SummerDaily17 (Fig. 8), the melt peak is Day 56 (26th July). What this suggests is twofold: first, that there is a lag of around 20 days for the full impacts of peak melt to feed through the entire subglacial system, including temporary storage and slow flow in the distributed sheet, and second, that one day of higher melt, i.e. a peak, is less important for building an extensive channelised drainage system than a sustained period of higher melt. The importance of storage is further exhibited by the strong correlation we find between it and surface melt – 0.67 for

SummerDaily12 and 0.77 for SummerDaily17 – indicating that much of the excess meltwater on high-melt days is impounded for a time, rather than transiting the subglacial drainage system. Note that, for both daily-forced runs, the maximum system extent occurs near the end of a period of sustained higher surface melt and is not replicated by similar shorter periods of higher melting that happen earlier or later in the melt season.

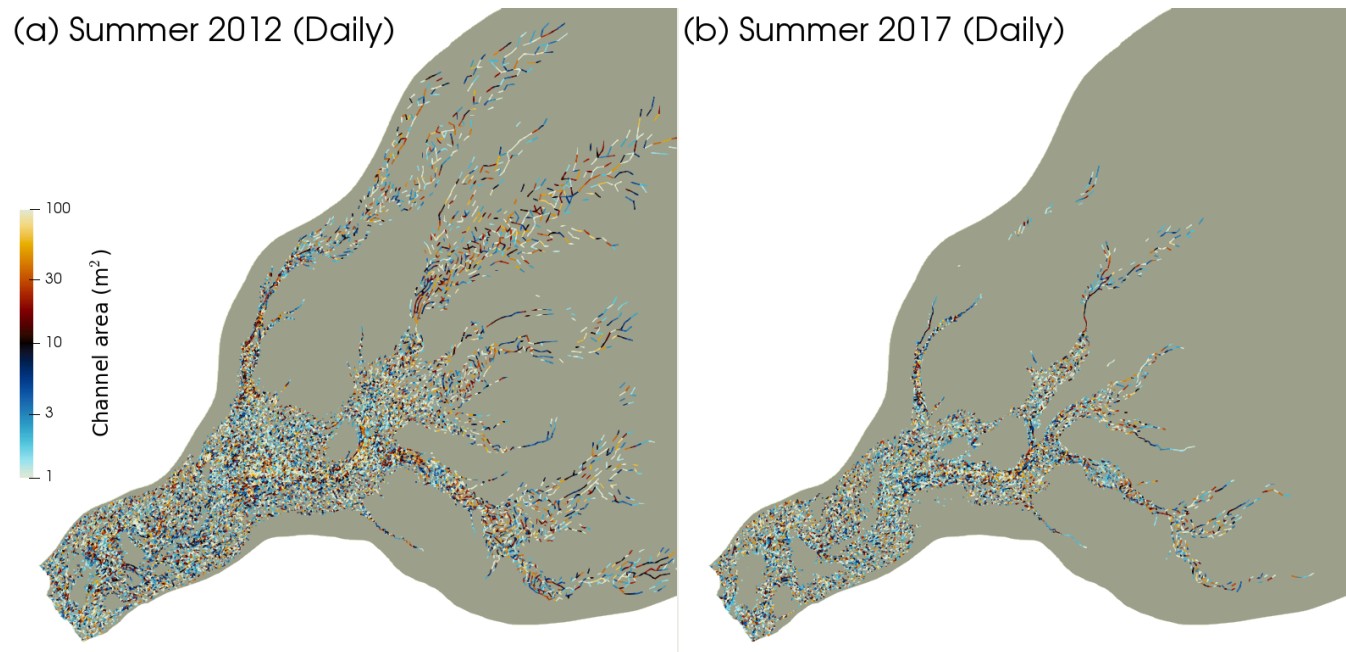

**Fig. 9 – Maximum extent of channelised subglacial drainage in (a) SummerDaily12 run on July 31st and (b) SummerDaily17 run on August 14th. It is instructive to compare these two panels with panels (c) and (d) of Fig. 5 to show the variability of the channelised system over a melt season.**

### 4.3. Summer plume activity

Turning to how these seasonal changes in the subglacial drainage system impact plume activity at the calving front, an interesting outcome is that the average melt rate in the SummerAverage17 model run is more than double that found in the Winter simulation, while the mean maximum melt rate is over six times higher (Table 3). Similarly, in SummerAverage12, the average melt rate is nearly four times greater than in Winter, whereas the mean maximum melt rate is nearly eight times higher (Table 3). Clearly, the much greater freshwater flux in either summer compared to winter is preferentially enhancing maximum melt rates compared to average melt rates. The explanation lies in the changing structure of the hydrological system: the greater degree of channelisation in summer leads to larger, more localised plumes at the expense of the more diffuse, primarily distributed-sheet-discharge-driven plume extending the length of the calving front. In other words, the extra water is preferentially concentrated by channels at a few points, rather than being spread out evenly over the entire width of the front. This is borne out by Fig. 4, where several substantial localised plumes are visible in summer (panels b-e), instead of a more uniform strengthening of the winter melting pattern (panel a) across the entire front. The larger, more localised plumes in

summer drive much more melting in their immediate vicinity, hence the higher modelled maximum melt rates in summer, but leave the remaining calving front comparatively less affected by plume-induced melting, reducing their impact on the modelled average rates. The latter is corroborated by Slater et al. (2015), with respect to the relative impacts of distributed and channelised drainage systems on plume melt rates. This pattern would also serve to promote calving by enhancing localised

concentrated melting from plumes, creating a more indented and less stable calving front, as posited by Todd et al. (2019) with regards to calving behaviour at Store. Todd et al. (2019) further suggest that this pattern could be enhanced by longer or warmer summers, a suggestion supported by our findings in this study of greater plume activity in the warm summer 2012 compared to the cool summer 2017 (Fig. 4).

This enhancement in plume melting is slightly reduced in both daily-forced runs (SummerDaily12 and SummerDaily17) (Fig.
4d, e), though much more noticeably with regard to the mean maximum than the average rate, which we relate to the greater variability of meltwater forcing reducing the activity and lifespan of the largest plumes compared to the average-forced runs (SummerAverage12 and SummerAverage17). Overall, though, the pattern of greater localised melt driven by channel formation remains strong in the daily-forced runs. Identifying whether more rapid, more focused channel-driven melting or slower, more diffuse distributed-sheet-driven melting is more important for calving and glacier dynamics is currently a subject

of debate and one we hope to investigate in future work, though, as described above, recent work by Todd et al. (2019) suggests the former, which promotes high localised melting and calving-front instability, is of greater importance. It is also important to note that our mean maximum plume melt rates for all summer simulations (Table 3) accord well with the observed summer melt rate at Store of $3.4\pm0.7$ m d$^{-1}$ from Chauché (2016), measured using side-scan sonar in summer 2012, and with other modelling studies for Greenlandic glaciers (Xu et al., 2013).

This slight reduction in concentration of melt in the largest plumes in the daily-forced runs also explains the very slight increase in total plume-induced melting (on the order of 2%) found compared to the average-forced runs, as the marginal favouring of the distributed sheet-driven plume spreads higher melt rates over a larger area. However, the difference is very small, and suggests that, if operating glacial hydrological models at longer temporal and/or larger spatial scales, averaged inputs yield similar outputs to daily-resolution data. Whether this remains the case in a fully-coupled simulation would be an interesting

target for future work.

The summer plume results also reinforce the point made in Sect. 4.1, above, about the importance of the depth of the grounding line for plume activity. There are many areas of strong plume melt towards the centre of the calving front (Fig. 4, Fig. 10), where subglacial discharge is quite low (Fig. 5, Fig. 6, Fig, 9), but the warmer, more saline water at the greater depths (>400 m) reached in this region of the front (Fig. 2) still allow high plume melting to occur without needing much meltwater input.

Conversely, despite higher meltwater discharges nearer the margins, the relatively shallow fjord depth and, therefore, colder, fresher ambient conditions (Fig. 2) limit the amount of melting the resulting plumes can achieve. From our model results, consequently, it is clear that the presence and location of warm, saline water in the fjord is equally important for generating plume melt as is sustained subglacial meltwater discharge, in line with buoyant plume theory (Jenkins, 2011; Slater et al., 2016).

A further possibility for validation is provided by the location of the plumes: visible plumes at Store have been observed persistently about 2 km in from the southern margin of the terminus (i.e. about one third in from the right of Fig. 4) and intermittently in the northern embayment (a similar distance in from the left of Fig. 4) (Ryan et al., 2015). Our model predicts the intermittent northern plumes well, but does not produce a persistent plume at the observed location on the southern half of

5 the terminus. Rather, the modelled plumes are more mobile and do not persistently occupy one location, with several hotspots of plume activity in the southern half of the terminus (Fig. 10). The reasons for this are considered in Sect. 4.5, below.

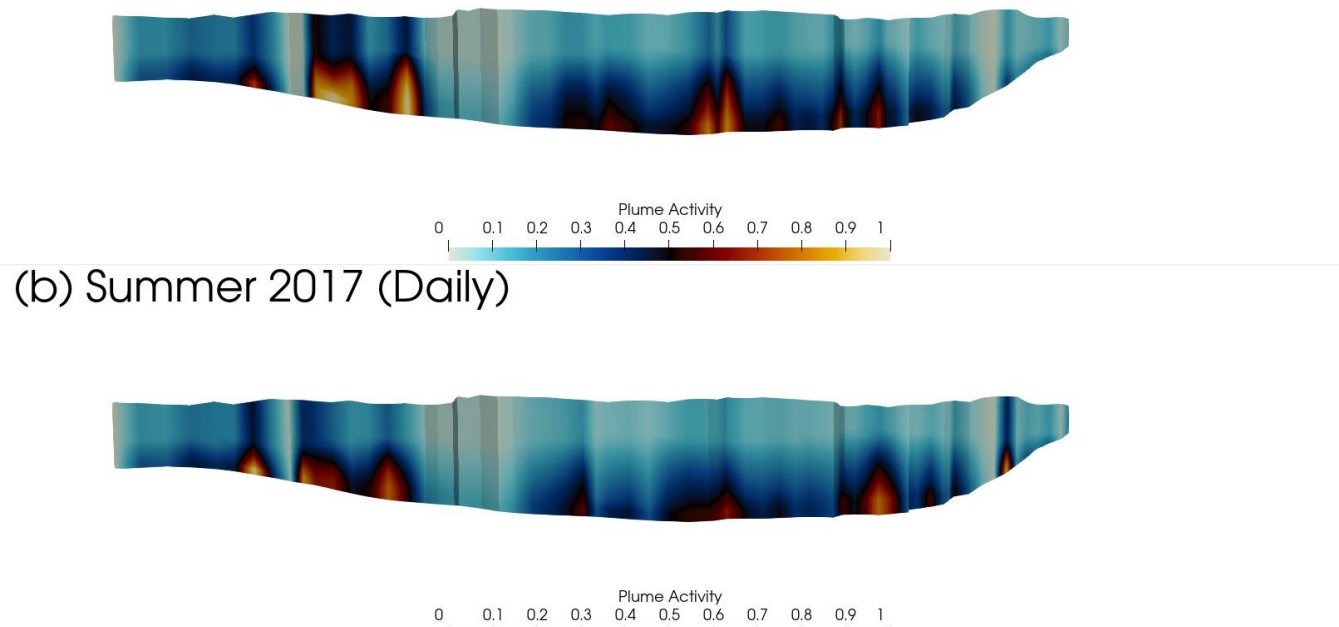

**Figure 10 -** **Heat map of plume activity in (a) SummerDaily12 and (b) SummerDaily17 simulations. Areas with a value of 1 show the highest mean plume melt rates across the entire length of the model run; areas with a value of 0 show no plume activity at any point.**

The relationship between plume activity and surface-melt variability is also critical to simple parameterisations of submarine melting. Many studies based on buoyant plume theory or high-resolution ocean modelling show a sublinear relationship between subglacial runoff and submarine melting, that is submarine melt rate is proportional to runoff raised to some power 0.25-0.9 (Jenkins, 2011; Slater et al., 2016; Xu et al., 2013). However, when considering surface melting, many studies assume a direct relationship between this and subglacial discharge, and, consequently plume melting (e.g. Carroll et al., 2016; Mankoff

et al., 2016; Stevens et al., 2016; Slater et al., 2019). A scatter plot of surface melting versus submarine melting for our SummerDaily17 simulation does not, though, show a strong relationship of this form (Fig. 11). Linear regression suggests surface melting explains only 21% of variability in plume melting (39% for 2012). We therefore propose that the structure of the subglacial drainage system and the associated water storage play a crucial role in mediating and smoothing water delivery

to the calving front, such that variation in plume activity is only partially relatable to peaks and troughs in surface meltwater production. This mediating role of the hydrological system appears in this case to obfuscate a simple relationship between surface and plume melt. This chimes with the important role assigned to subglacial and englacial water storage by the outcomes of the SHMIP process (De Fleurian et al., 2018).

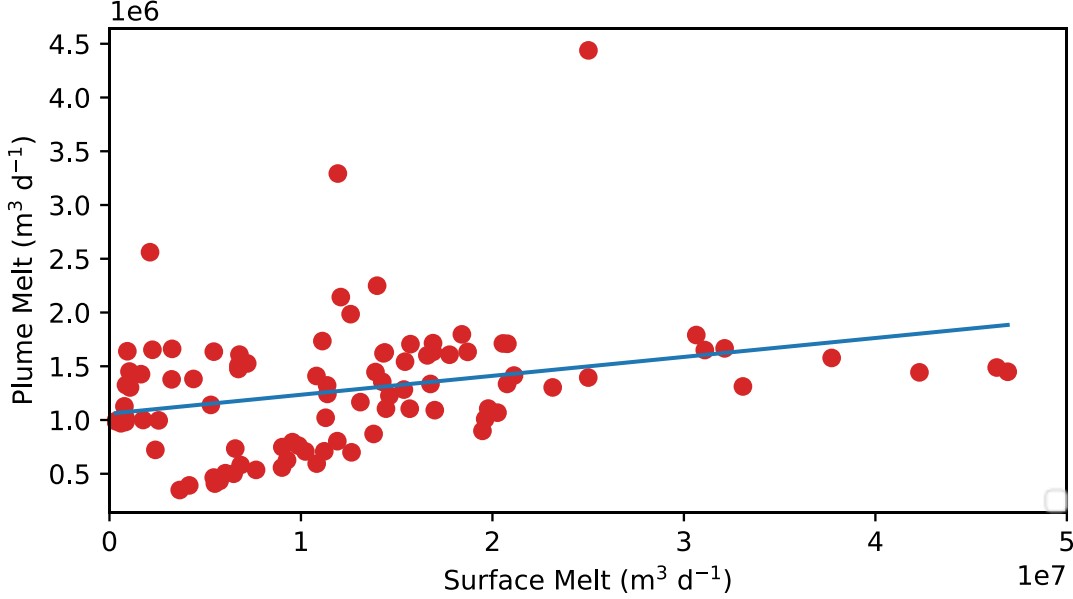

**Fig. 11 – Scatter plot of surface melt versus plume melt for SummerDaily17 run, showing low correlation. The line of best fit is shown in blue.**

## 4.4. Implications for glacier dynamics

Although daily-scale changes in surface melt are poorly correlated with plume activity at the calving front, they do show a

10   close relationship with other aspects of the hydrological system. This is shown by Fig. 12, which displays the domain-averaged water pressure versus the domain-averaged surface melt for the SummerDaily12 and SummerDaily17 runs. Peaks in surface melt are lagged by peaks in water pressure, usually by one day of model time, throughout the simulation. The correlation coefficient is 0.67 for SummerDaily12 and 0.77 for SummerDaily17, confirming the strength of this relationship.

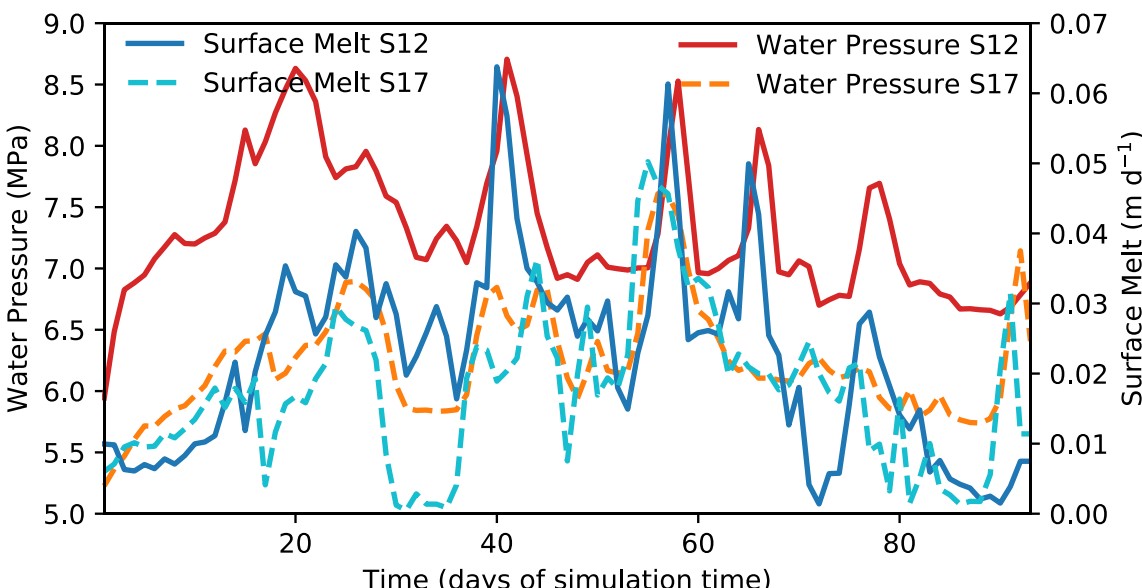

**Fig. 12 – Time series showing comparison between average water pressure (left axis) and surface melt (right axis) for SummerDaily12 (solid lines) and SummerDaily17 (dashed lines) model runs. Note the correlation between surface melt and water pressure.**

One final point of interest is that, despite the greatly increased channelisation evident in the SummerAverage12 and SummerAverage17 model runs compared to the Winter run, modelled effective pressures decrease (i.e. modelled water pressures increase) in summer compared to winter, contrary to expectation (e.g. Meierbachtol et al., 2013). Given that we are not coupling the hydrology to the ice flow in this study, we will not address the implications for the flow of Store, save to make two brief points. The first is that modelled effective pressures are higher (i.e. modelled water pressures are lower) in the SummerDaily12 and SummerDaily17 runs than in the SummerAverage12 and SummerAverage17 runs, which suggests that the much lower effective pressures found in the latter runs are partly an artefact of the seasonally averaged surface-melt forcing. The second is that, looking at the maximum extent of the hydrological system in the SummerDaily12 run, we find lower effective pressures (i.e. higher water pressures) than for the maximum extent of the SummerDaily17 run, despite having a larger and more extensive low-pressure channel network. This perhaps indicates that even the record levels of melt in 2012 were unable to generate a low-pressure channel system of sufficient extent to evacuate all the water efficiently. If there is a melt threshold at which a fully efficient subglacial drainage system can develop at Store, it must therefore likely be at a level of melt not yet reached. Investigating what effect these hydrological changes have on ice dynamics at Store will be a focus of future work

### 4.5. Limitations and future work

One limitation of this study is the lack of two-way coupling between ice flow and subglacial hydrology, the fixed ice geometry and absence of calving processes. This simplification was used to allow us to focus purely on the evolution of the subglacial

hydrological system under different forcings, inside a state representative of the long-term state of Store, and greatly reduced the computational cost of the study. Given four decades of stability of Store (Rignot et al., 2015), we also feel this is a reasonable simplification to make. Consequently, we have focused our discussion on the structure and behaviour of the hydrological system, rather than speculating as to the likely impacts of this behaviour on ice flow, which would require a fully

coupled study to investigate.

As can be seen from the description in Sect. 2.3., the hydrological model results are also ultimately dependent on the mesh. We consider that the fine resolution of the mesh throughout the area of high water flux obviates this problem. Fig. 13 further shows a comparison between the results from the hydrological model for the SummerAverage12 simulation – Fig. 13a shows the results on the standard mesh used for all simulations and described in Sect. 2.3.; Fig. 13b shows the same results calculated

on a mesh of constant 500 m resolution. As can be clearly observed, the overall pattern of the channel network remains similar between the two meshes, though, evidently, the detail of which individual channel segments are most important varies, as there are simply far fewer flow paths available on the coarser mesh. Overall, this gives us confidence that the pattern of our findings is robust, though it does caution against over-interpreting the fine details. This is further supported by the mesh dependency analysis undertaken by Werder et al. (2013) for GlaDS, which shows little variation in results in the presence of realistic

topography.

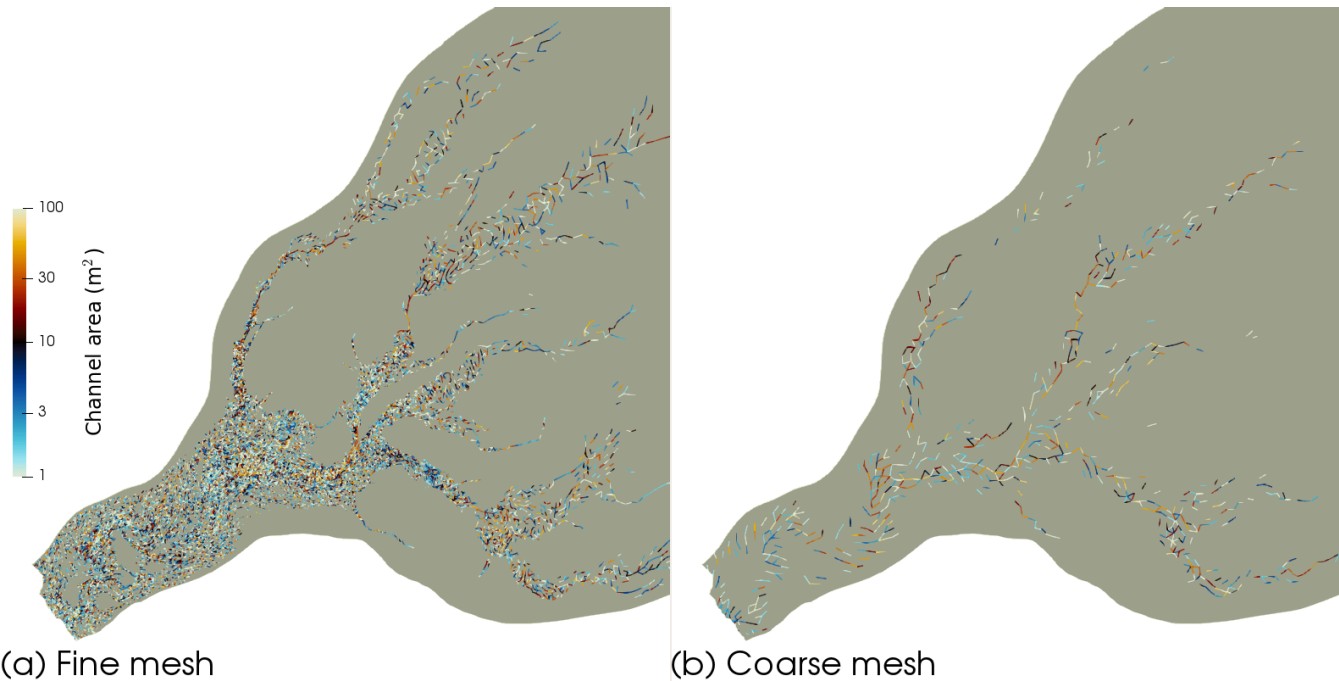

(a) Fine mesh            (b) Coarse mesh

**Fig. 13 – Comparison of hydrological model results for different mesh resolutions. (a) The standard, fine mesh described in Sect. 2.3.; (b) a coarser, 500 m-resolution mesh. Note that the overall layout of the channel network is virtually identical between the two.**

Another model limitation is the simplified grounding line, which impacts our plume results. In reality, the area of high plume

activity at the deepest part of the calving front (Fig. 4, Fig. 10) might form one single plume that would reach the surface, an

effect that the model would likely reproduce with a more realistic grounding line. The model also shows a tendency for the plume activity in that region to migrate towards the centre of the calving front over the course of the simulation, which is also likely due to the simplified grounding line used in this study. Both these effects occur because the observed grounding line for the deeper, southern side of the terminus is a kilometre or so inland from the calving front. Even though it is small, this floating section could interrupt the water flow from the southern part of the terminus towards the centre (Fig. 5) and therefore shift the area of modelled enhanced plume activity back to its observed position and concentrate the discharge more stably. With the simplified setup in this study, the hydrological system oscillates between several potential stable states in the region of low hydraulic gradient immediately behind the calving front, similar to behaviour inferred from seismic observations on similar regions in real glaciers (Vore et al., 2019). We have therefore generally confined our discussion of plume melt to average values, which are less dependent on any one specific pattern of plume activity, rather than over-interpreting such patterns. Including a full representation of the grounding line to mitigate these issues and be able to realistically investigate how plume discharge locations move over time will be part of our future work on this model.

Finally, the plume model relies on several poorly-known parameters, which result in a high degree of uncertainty around the resulting melt rates. In particular, the heat and salt transfer coefficients, which determine the rate at which heat is transferred from the ocean to the ice, are very poorly constrained. This results from the extreme difficulty of directly observing submarine melt rate at tidewater glaciers simultaneously with all of the other factors affecting submarine melting, such as fjord conditions and circulation, and grounding-line subglacial hydrology. Until better observations are available to place constraints on models, the absolute values of melt rates in studies such as this should be viewed with caution. On the other hand, relative comparisons of melting, for example from location to location on a calving front (Fig. 4) or between two seasons or time periods (Table 3) are more robust with regards to this uncertainty. It is also our hope that models such as ours will help to reduce these prevalent uncertainties on melting through improving understanding of near-terminus subglacial hydrology. A similar problem applies to the parameters used for GlaDS – observational difficulties mean they are currently poorly constrained, but we hope to improve this by undertaking a full validation exercise, through comparison with an independently derived dataset of calving events at Store, upon the completion of development of a coupled ice-hydrology-plume-calving model, which is the focus of our future work.

## 5. CONCLUSION

We present the first coupled hydrology-plume model applied to a tidewater glacier in Greenland, allowing us to investigate aspects of the subglacial hydrology of Store Glacier critical to ice dynamics and calving-front melting that are poorly constrained by existing observations and models. We demonstrate that the implementation of the GlaDS hydrological model within the Elmer/Ice modelling suite shows promise in realistically recreating the observed behaviour of the subglacial drainage system of Store (Chauché, 2016; Doyle et al., 2018; Young et al., 2019), giving us greater confidence in its use as a predictive tool.

By modelling the seasonal changes in the subglacial hydrology of Store, we explore how discharge drives convective plumes that melt the submerged portion of the terminus. We find an active subglacial drainage system, with small channels and a distributed sheet extending up to 45 km inland in winter, which drives substantial plume activity across the calving front, with localised melt rates of up to 1.1 m d$^{-1}$. This means the freshwater flux is non-zero in winter, at 5.96 m$^3$ s$^{-1}$, which contrasts with

assumptions of zero winter freshwater flux at tidewater glaciers in previous work. In summer, when surface melt is incorporated as an input to the drainage system, the drainage system extends up to 65 km inland, the distance inland that surface melting occurs, though significant channelisation only reaches up to 55 km. The more-developed channel system intensifies the activity of large plumes at the front, thereby raising the maximum rate of plume-induced melting to 12.6 m d$^{-1}$. However, the concentration of water in fewer larger channels also leaves a large portion of the calving front exposed to only

a weak plume, such that average plume melt rates increase by a much smaller factor compared to winter.

Overall we find plume melting to increase the freshwater flux into the fjord by 58% in winter, when the basal drainage system predominantly carries water produced by friction at the bed. In summer, when the basal drainage system also carries surface melt, plume melting increases the freshwater flux by only about 5%, on average, although it represents a higher absolute value. Overall, we find the freshwater flux to be 9.44 m$^3$ s$^{-1}$ in winter, with contributions of 42% and 58% from basal meltwater

production and plume-induced melting, respectively. In summer 2012, the contributions were 95% from surface and basal meltwater production and 5% from plume melting; and in 2017 91% and 9%, respectively.

We also demonstrate that peaks in surface melt are not well-correlated with peaks in plume melt, nor are they the dominant force in determining the maximum extent of the subglacial hydrological system, which is instead defined by longer periods of sustained melting. Finally, we show that basal water pressures in our model were higher during the record warm summer in

2012 compared to 2017 when surface conditions were close to the decadal average. Modelled effective pressures therefore suggest that the high melt inputs in 2012 did not form a fully efficient subglacial drainage system even though the latter extended 55 km inland. This indicates that channel formation may not fully negate the lubricating effects of high melt on ice flow. Future work will aim to couple ice flow and calving with the hydrology in order to simulate the dynamic effects of changes in water inputs and plume melting.

**AUTHOR CONTRIBUTION**

SC, PC and JT designed the experiments. SC also developed the model code and executed the experiments with contributions from JT, PC and DS. NC provided hydrographic data. SC analysed the model outputs and wrote the manuscript, with significant contributions from all co-authors.

**COMPETING INTERESTS**

The authors declare that they have no conflict of interest.

## ACKNOWLEDGEMENTS

The research was supported by the European Research Council under the European Union's Horizon 2020 Research and Innovation Programme. The work is an output from grant agreement 683043 (RESPONDER). SJC also acknowledges financial assistance in the form of a studentship (RG87486) from the Natural Environment Research Council. The authors thank Marion Bougamont, Tom Cowton and Thomas Zwinger for productive discussions, as well as Olivier Gagliardini for support with numerical and Brice Noël who provided the RACMO data. DEMs provided by the Polar Geospatial Center under NSF-OPP awards 1043681, 1559691, and 1542736. DS acknowledges funding from NSF OPP-1418256.

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
