# Peer review of "Coupled modelling of subglacial hydrology and calving-front melting at Store Glacier, west Greenland"

_The Cryosphere, 2019_

## Referee Comment (RC1) · Douglas Benn (Referee) · 2 Jul 2019

This paper represents a significant advance in our ability to analyse the basal drainage systems of tidewater glaciers. It is undoubtedly suitable for publication in The Cryosphere, and needs very little revision. It is very well written, has a clear and logical structure, and presents key results clearly and effectively.

At present, the paper is very tightly focused on the results, and there is some risk that it could be viewed simply as an interesting case study. As pointed out by the authors in the Introduction, the hydrology of tidewater glaciers has important consequences for rates of ice flow & the impact of meltwater plumes, and it would be worth linking

back more strongly to these themes in the Discussion. I therefore recommend that the authors make a few additions to the text to highlight the wider significance of the results and increase the impact of the work.

A key area in this regard is the prospect of including basal hydrology in prognostic models of tidewater glaciers. At present, prognostic models typically employ simple parameterizations of frontal ablation and basal friction. The hydrological model presented in this paper opens up the exciting possibility of modelling both frontal ablation and ice dynamics from first principles, making models much more adaptable to changing environmental and glaciological conditions. The present paper is, of course, still some steps away from this, but the results can already be used to flag up some important issues. In particular, the authors can shed light on the importance of seasonal and diurnal fluctuations, and comment on appropriate temporal averaging periods in simulations.

The importance of the contrast between winter and summer conditions, and of inter-annual variability, is made clear in the Winter Baseline and SummerAverage12 & SummerAverage17 runs. However, the wider significance of the Daily runs could be usefully explored in more detail. The authors limit presentation of the Daily runs to describing the 'maximum' and 'end' states. These results are of course interesting, but the differences between these states and the SummerAverage runs are hardly surprising. It would be of potentially greater interest to see how the seasonally integrated results of the Daily runs compare with the SummerAverage runs. In other words, do the SummerAverage simulations yield good approximations of the integrated Daily outputs? Do they predict the same mean plume locations and overall frontal melt totals? Or does system non-linearity mean that the SummerAverage and integrated Daily results yield different answers? A comparison between the SummerAverage and integrated Daily results could shed important light on the appropriate timescale for modelling frontal ablation and dynamic processes. Do we need to model these systems with daily resolution? Or are seasonal (or annual) averages sufficient? How can models help us

bridge between these timescales?

Another important issue is that of model validation. The model results are described clearly and in detail, and are of obvious interest in themselves. But the real power of the results lies in what they might tell us about what is going on beneath actual tidewater glaciers. Some useful comparisons between model output and independent data are made in the present text (e.g. comparing modelled winter frontal melt rates with the results of Chauché (p. 18); and comparison of the modelled basal water pressures and channel extent with the work of Doyle et al. and Young et al. (p. 20)). More should be made of these comparisons, and I suggest expanding these sections to provide more detail on the success (or limitations) of the model. Current opportunities for additional model validation are somewhat limited, but some should be possible. Are the predicted locations of plumes consistent with observations? Comparison of model output with time series of ice velocity might powerfully validate the water pressure results, but this would require additional model runs (e.g. to overlap with the TerraSAR-X data for 2014-2015 reported in Young et al., 2019). I don't expect the authors to undertake extra work for the current paper, but the possibility of this strand of model validation should be mentioned, and certainly considered for the future.

Specific comments:

Abstract, line 15. "In winter, we find channels over 1 m2 in area occurring up to 5 km inland, which shows that the common inference of zero winter freshwater flux is invalid" You could have non-zero flux without channels, so this statement does not follow logically. Change to something like: "We show that the common assumption of zero winter freshwater flux is invalid, and find channels over 1 m2 in area occurring up to 5 km inland."

p. 6, 15: regarding the assumption that "surface melt travels straight to the bed at the point of production", it is worth noting that this is reasonable on a heavily crevassed glacier.

[Figure]

p. 11, 14: it would be useful to cite a typical thickness of the sheet, and a threshold value when the sheet begins to transition to small channels.

Section 3.2: Did the modelled drainage system reach steady state by the end of the 3 month simulations?

Additionally, more detail is needed in the caption to Fig. 4: Panels b & c: are these pictures of the end of the simulation? Panels d & e: what days of the simulation are shown? Are these for 'maximum' conditions?

Section 3.3 and Table 3: See comments above on presenting results from integrated Daily runs. How do the overall mean values of the Daily runs compare with the Average runs? Can simulations based on seasonal averages yield good approximations of seasonal average outputs (e.g. location of plumes and melt-undercutting totals), or does system non-linearity mean that daily simulations are necessary? Add data to Table 3 and present results in Section 3.3, plus appropriate discussion in Section 4.2 & 4.3.

p. 18, 18: Here you compare the model output with the results of Chauché (2016). Since this source is an unpublished PhD thesis, you need to provide more context here. What methods were used by Chauché? What were the associated errors? Are the current results more or less reliable than those of Chauché?

p. 19, 4: The deep fjord water is not 'subtropical'. Use 'warm Atlantic Water' instead.

---

## Author Comment (AC1) · 5 Jul 2019

Dear Professor Benn

Thank you for your helpful comments on our paper. We agree that presenting the integrated plume melt and position results across the different simulations would be an interesting addition to the paper and allow us to make it more effective, as would tying the results back to the observations in a more comprehensive manner. We are also pleased you pointed out the importance of validation – we intend to address this issue in future work, but accept that this could be made clearer in the present paper.

[Figure]

Thanks again for taking the time to review our paper

Samuel

---

## Referee Comment (RC2) · Anonymous Referee #2 · 12 Jul 2019

This is a potentially interesting paper describing a modelling study of sub-glacial water flow beneath Store Glacier and its impact on plume activity at the calving front. The subject matter lies clearly within the scope of the Cryosphere, and the study is novel in its combination of sub-glacial hydrology and buoyant plume theory. The links have been discussed extensively in the literature, but have not previously been quantitatively linked in the way that they have in this study.

Despite its clear merits, I think that some further work is required on the manuscript before it can be accepted for publication. The main weakness is that the scientific motivation for the study is never clearly presented or addressed. This issue is appar-

ent even in the title, which focuses on the technical achievement of running the model rather than the findings of the experiments. Throughout, the presentation focuses on detailed numerical outputs (often quoted to 3 significant figures), but nowhere is there an assessment of how reliable those numbers are, either through a detailed comparison with observation or a study of model sensitivity to parameter choice. So the reader (this one at least) is left puzzled as to what the key messages are. The qualitative results that surface melting drives a more active sub-glacial hydrological system, that there is a delay as the increased supply of water works its way through the system to emerge at the grounding line, and that increased discharge there drives higher melting along the calving front, are all well known. If my interest is in the details of Store Glacier during the two summers studied, I could potentially use the quantitative results, but then I really need that missing assessment of how well-constrained the numbers are. If my interest is in the broader subject of sub-glacial drainage and its role in stimulating melting along the calving front, how do these numerical results help me? The authors really need to think through what they want readers to take from this study. If it is nothing more than the demonstration of a working model, perhaps the paper should be recast as a more thorough description and critical assessment of that model. Such a paper might be more appropriate for a journal such as Geoscientific Model Development.

More detailed comments:

Page 1, line 16: Here and in a number of other places the authors comment on the "common assumption" that sub-glacial discharge at a tidewater glacier terminus falls to zero in winter. I was not aware of that being a common assumption. Indeed plume models tend to require a non-zero outflow as an initial condition, so all the studies I know of that are based on plume theory have a background flow in winter by default. Perhaps the authors could provide specific citations to the studies that assume zero outflow?

Page 6, line 5: I'm not sure I follow these equations. Are they specific to the calving

[Figure]

front? In that case shouldn't the density in (3) be the seawater density?

Table 1: How were these parameter values chosen? How sensitive are your results to those choices?

Figure 2: I think this figure would be more informative if one panel showed temperature in winter and summer (on the same scale), while the other showed salinity in winter and summer.

Page 12, lines 1-4: I don't agree with this statement. If the bed topography controlled the hydraulic potential gradient, the water would pool in the deepest part of the bed.

Results: The text in this section could be reduced significantly. Much of it presents information that is readily available in Table 3. The reader should simply be told why those numbers are of interest. A detailed quantification of their relative sizes is unnecessary.

Figure 4: You never say whether the "Daily" results are from the end of the summer, or the time of peak meltwater input, or perhaps the time of peak meltwater discharge.

Page 18-19: This is one of the few places where any comparison with observation is made. The results do not compare very well: the grounding line flux is at the low end, but the calving front melt rate is an order of magnitude smaller. However, the observational numbers appear to be poorly constrained. When were the observations made? Were there no analogous measurements made in summer? A little more discussion of is called for.

Page 20, lines 16-19: Another very cursory comparison with observation. Is the nature of the drainage system the only result that can be used to validate the model? It would help to plot the location of the observations in Figure 5.

**Page 22, lines 5-6: Aren't these also results that could be compared with observation? Are sediment-laden plumes seen in satellite imagery obtained in the summer months? How do the times and locations of their appearance compare with your model

results?

Page 22, lines 25-27: The sub-linear relationships discussed are between sub-glacial discharge and melting. I am not aware of any study that has related surface melting (except in the average sense) directly to sub-glacial melting for the simple reason that there is a delay before surface melt emerges at the grounding line. Your Figure 10 does not take this into account, so I'm not sure what the point of showing it is. If you related discharge to melt, I assume you would see the same sort of relationship as others, since it would be a product of your plume model. More useful would be if you could show some sort of correlation between surface and submarine melt with some simple processing (maybe smoothing and a lag) applied to the surface melt signal. That would be a step towards a simple parameterisation of the overall impact of the sub-glacial hydrological network.

Figure 12: You show the differences between the results obtained with a coarse and a fine model grid, but has the solution converged on the finer mesh, or would further refinement give different results again?

Page 25-26: There is a suggestion here that the agreement between modelled and observed plume locations (see ** above) is poor. It would be more honest to actually show this comparison, especially if the discrepancy can be explained. But if explanation is the "unrealistic" grounding line, why not use a more realistic one?

Page 26, lines 13-14: There is a mention of parameter uncertainty in the plume models, but no mention here or anywhere else about parameter uncertainty in the sub-glacial hydrology model.

Page 26, lines 22-24: This claim is really not supported by the paper. There is very little comparison with observation, and the comparisons that are made show significant discrepancies. That is a major issue with the paper.

Page 27, lines 3-8: Are these potentially testable results? If you have measurements

of water properties in the fjord, can you diagnose the relative inputs of sub-glacial meltwater versus that produced by melting of the calving front?
* * *

---

## Author Comment (AC2) · 16 Jul 2019

Dear Reviewer

Thank you for taking the time to read our manuscript. We agree with your comments about needing to improve our comparisons to observations and we accept that we could do more to clarify the rationale and targeting of the paper.

As regards the choice of parameters and the model sensitivity to these, the values were taken from previously published work (Gagliardini and Werder, 2018) and sensitivity analysis of the GlaDS model was undertaken by Werder et al. (2013). We accept

that this is not necessarily directly applicable to our model domain, but a full sensitivity analysis seems both unnecessary and outside the scope of this paper. We are also aware that a full validation of the model needs to be undertaken, and this will be the subject of future work, but we also felt that it was not suitable for this paper, as the model presented here is not the fully-coupled version we are working towards. We feel that the results from this idealised uncoupled model are interesting enough, however, to justify publication at this stage.

Once again, we thank you for your comments, which will allow us to improve the paper.

---

## Author Comment (AC3) · 6 Aug 2019

To round off the discussion on this paper, we have listed the specific comments from both referees, as well as any general comments not covered by one or more specific comments, below, and provided our response to each one. Each block of text consists of an editorial comment followed immediately by our response to it.

Some useful comparisons between model output and independent data are made in the present text (e.g. comparing modelled winter frontal melt rates with the results of Chauché (p. 18); and comparison of the modelled basal water pressures and channel extent with the work of Doyle et al. and Young et al. (p. 20)). More should be made

of these comparisons, and I suggest expanding these sections to provide more detail on the success (or limitations) of the model. Current opportunities for additional model validation are somewhat limited, but some should be possible. Are the predicted locations of plumes consistent with observations? Comparison of model output with time series of ice velocity might powerfully validate the water pressure results, but this would require additional model runs (e.g. to overlap with the TerraSAR-X data for 2014-2015 reported in Young et al., 2019). I don't expect the authors to undertake extra work for the current paper, but the possibility of this strand of model validation should be mentioned, and certainly considered for the future. We will add text to our comparisons to observations to make them more meaningful, and will add the study site location for Doyle et al. (2018) and Young et al. (2019) to Figures 5d and 6d to assist the reader. We will also add text discussing the match between modelled and observed plume locations and give more detail on our plans for model validation.

Abstract, line 15. "In winter, we find channels over 1 m2 in area occurring up to 5 km inland, which shows that the common inference of zero winter freshwater flux is invalid" You could have non-zero flux without channels, so this statement does not follow logically. Change to something like: "We show that the common assumption of zero winter freshwater flux is invalid, and find channels over 1 m2 in area occurring up to 5 km inland." Wording will be changed as suggested.

p. 6, 15: regarding the assumption that "surface melt travels straight to the bed at the point of production", it is worth noting that this is reasonable on a heavily crevassed glacier. Words will be added to make this point.

p. 11, 14: it would be useful to cite a typical thickness of the sheet, and a threshold value when the sheet begins to transition to small channels. Words will be added to provide some information on typical sheet thicknesses. We will not, however, quote a value for when the transition to channels happens as this transition is not simply dependent on the sheet thickness and occurs at different values in different parts of the glacier.

Section 3.2: Did the modelled drainage system reach steady state by the end of the 3 month simulations? Additionally, more detail is needed in the caption to Fig. 4: Panels b & c: are these pictures of the end of the simulation? Panels d & e: what days of the simulation are shown? Are these for 'maximum' conditions? We will clarify that the simulations did not reach a steady state and will expand the caption for Figure 4 to state more clearly what the panels are showing..

Section 3.3 and Table 3: See comments above on presenting results from integrated Daily runs. How do the overall mean values of the Daily runs compare with the Average runs? Can simulations based on seasonal averages yield good approximations of seasonal average outputs (e.g. location of plumes and melt-undercutting totals), or does system non-linearity mean that daily simulations are necessary? Add data to Table 3 and present results in Section 3.3, plus appropriate discussion in Section 4.2 & 4.3. We will add a row to Table 3 detailing integrated plume melt across all simulations and include text in Sections 3.3 and 4.3 presenting and discussing this.

p. 18, 18: Here you compare the model output with the results of Chauché (2016). Since this source is an unpublished PhD thesis, you need to provide more context here. What methods were used by Chauché? What were the associated errors? Are the current results more or less reliable than those of Chauché? Text will be added to provide more context on Chauché (2016).

p. 19, 4: The deep fjord water is not 'subtropical'. Use 'warm Atlantic Water' instead. Wording will be changed as suggested.

The main weakness is that the scientific motivation for the study is never clearly presented or addressed. By extending our comparison to observations throughout the paper, we believe we will address this point by making the paper a more useful, constrained modelling study of Store Glacier. We also highlight the text in Section 1, which clearly positions the paper within the relevant theoretical context.

Page 1, line 16: Here and in a number of other places the authors comment on the

"common assumption" that sub-glacial discharge at a tidewater glacier terminus falls to zero in winter. I was not aware of that being a common assumption. Indeed plume models tend to require a non-zero outflow as an initial condition, so all the studies I know of that are based on plume theory have a background flow in winter by default. Perhaps the authors could provide specific citations to the studies that assume zero outflow? We will add some relevant references.

Page 6, line 5: I'm not sure I follow these equations. Are they specific to the calving front? In that case shouldn't the density in (3) be the seawater density? Yes, these equations are specific to the boundary condition at the calving front, be this in a lake or a fjord. In this case, therefore, the relevant density would be of seawater, but the equations do not require this – the density term is just the density of whatever water the calving front is in. We will add text to clarify this.

Table 1: How were these parameter values chosen? How sensitive are your results to those choices? The choosing of the parameter values is explained in the text preceding Table 1 (p.6, line 18ff). The values were taken from previously published work (Gagliardini and Werder, 2018) and sensitivity analysis of the GlaDS model was undertaken by Werder et al. (2013). We accept that this is not necessarily directly applicable to our model domain, but a full sensitivity analysis seems both unnecessary and outside the scope of this paper.

Figure 2: I think this figure would be more informative if one panel showed temperature in winter and summer (on the same scale), while the other showed salinity in winter and summer. Will be changed as suggested.

Page 12, lines 1-4: I don't agree with this statement. If the bed topography controlled the hydraulic potential gradient, the water would pool in the deepest part of the bed. We say that the hydraulic potential gradient is mainly controlled by the bed topography, with flow paths following the deeper parts of the bed. We agree that if we were stating that the bed topography were the sole control on the hydraulic potential gradient, we

would be in error, but that is not the claim we are making.

Results: The text in this section could be reduced significantly. Much of it presents information that is readily available in Table 3. The reader should simply be told why those numbers are of interest. A detailed quantification of their relative sizes is unnecessary. We understand the point being made here and will consider the best way of addressing this going forwards.

Figure 4: You never say whether the "Daily" results are from the end of the summer, or the time of peak meltwater input, or perhaps the time of peak meltwater discharge. See response to earlier referee comment on Section 3.2, above.

Page 18-19: This is one of the few places where any comparison with observation is made. The results do not compare very well: the grounding line flux is at the low end, but the calving front melt rate is an order of magnitude smaller. However, the observational numbers appear to be poorly constrained. When were the observations made? Were there no analogous measurements made in summer? A little more discussion of is called for. We will add further discussion of the context surrounding the winter observations and some text to Section 4.3 to deal with the measurements made in summer. These were not analogous, being derived from side-scan sonar rather than CTD and ADCP data fed through a model, but offer a useful additional constraint.

Page 20, lines 16-19: Another very cursory comparison with observation. Is the nature of the drainage system the only result that can be used to validate the model? It would help to plot the location of the observations in Figure 5. The location of the observations will be plotted in Figures 5d and 6d and the comparison to observations will be expanded.

\*\*Page 22, lines 5-6: Aren't these also results that could be compared with observation? Are sediment-laden plumes seen in satellite imagery obtained in the summer months? How do the times and locations of their appearance compare with your model results? We will add some text comparing the modelled and observed locations of

plumes.

Page 22, lines 25-27: The sub-linear relationships discussed are between sub-glacial discharge and melting. I am not aware of any study that has related surface melting (except in the average sense) directly to sub-glacial melting for the simple reason that there is a delay before surface melt emerges at the grounding line. Your Figure 10 does not take this into account, so I'm not sure what the point of showing it is. If you related discharge to melt, I assume you would see the same sort of relationship as others, since it would be a product of your plume model. More useful would be if you could show some sort of correlation between surface and submarine melt with some simple processing (maybe smoothing and a lag) applied to the surface melt signal. That would be a step towards a simple parameterisation of the overall impact of the sub-glacial hydrological network. On further consideration, we agree that Figure 10 is not showing anything particularly valuable, so we will remove it and rewrite the accompanying paragraph to refer to Figures 7 and 8. It is sufficiently clear from Figures 7 and 8 that there is very little, if any, relationship between surface and plume melt, even if lags or smoothing were applied, so we have decided to not pursue this further.

Figure 12: You show the differences between the results obtained with a coarse and a fine model grid, but has the solution converged on the finer mesh, or would further refinement give different results again? No, the model has not converged on either mesh. In both cases, the end point of the run is shown and, in both cases, the channel network was still growing. We tried several different mesh resolutions in the initial work for this paper and a finer mesh resolution than the one eventually chosen both significantly increases model run time and generates numerical instabilities that crash the model, so was not pursued further, being impractical.

Page 25-26: There is a suggestion here that the agreement between modelled and observed plume locations (see ** above) is poor. It would be more honest to actually show this comparison, especially if the discrepancy can be explained. But if explanation is the "unrealistic" grounding line, why not use a more realistic one? We will expand our

discussion of the contrasts between observed and modelled plume locations in Sect. 4.3 to give more detail on the mismatch; given the relatively simple nature of this, an additional figure is unnecessary. We agree that using a more realistic grounding line would be ideal, but it is not a straightforward change to implement independently of the calving code we are currently working to integrate with the model, hence our decision to omit doing so for this study.

Page 26, lines 13-14: There is a mention of parameter uncertainty in the plume models, but no mention here or anywhere else about parameter uncertainty in the sub-glacial hydrology model. We will add a reference to uncertainty in the GlaDS parameters and discuss our planned solution of this by conducting a full validation exercise as part of future work.

Page 26, lines 22-24: This claim is really not supported by the paper. There is very little comparison with observation, and the comparisons that are made show significant discrepancies. That is a major issue with the paper. When we have expanded the comparisons to observations made throughout the paper, we will consider whether this statement remains valid. The model is by no means perfect, but its current failings are expected based on its simplified state. We acknowledge the lack of a full validation exercise undertaken as part of this study, but re-emphasise that this is something we intend to undertake, and subsequently publish, with the fully coupled version of the model.

Page 27, lines 3-8: Are these potentially testable results? If you have measurements of water properties in the fjord, can you diagnose the relative inputs of sub-glacial meltwater versus that produced by melting of the calving front? No, the measurements available do not allow this to be done.

---

## Author Response (AR1)

**Authors' Response to Referee Comments for: Coupled modelling of subglacial hydrology and calving-front melting at Store Glacier, West Greenland**

We would like to thank both reviewers for their comments on the paper. As a result we have expanded our comparisons of model results to observations and clarified the motivation behind the paper, as well as addressing all the specific comments as further detailed below. We believe this has strengthened the study under consideration by making the model's successes and failings clearer, and by more clearly positioning the paper within the wider literature.

Editorial Comments

Our response (page and line numbers refer to tracked-changes document below)

Reviewer 1

Some useful comparisons between model output and independent data are made in the present text (e.g. comparing modelled winter frontal melt rates with the results of Chauché (p. 18); and comparison of the modelled basal water pressures and channel extent with the work of Doyle et al. and Young et al. (p. 20)). More should be made of these comparisons, and I suggest expanding these sections to provide more detail on the success (or limitations) of the model. Current opportunities for additional model validation are somewhat limited, but some should be possible. Are the predicted locations of plumes consistent with observations? Comparison of model output with time series of ice velocity might powerfully validate the water pressure results, but this would require additional model runs (e.g. to overlap with the TerraSAR-X data for 2014-2015 reported in Young et al., 2019). I don't expect the authors to undertake extra work for the current paper, but the possibility of this strand of model validation should be mentioned, and certainly considered for the future.

We have added text to our comparisons to observations (p.31, lines 18-20; p.32, line 1; p.33, lines 22-29; p.36, lines 12-15) to make them more meaningful, and have added the study site location for Doyle et al. (2018) and Young et al. (2019) to Figures 5d and 6d (pp.25, 27) to assist the reader. We have also added text discussing the (mis)match between modelled and observed plume locations (p.36, lines 22-27) and redrawn Figure 4 (p. 23) to better show this. We have additionally given more detail on our plans for model validation (p.40, lines 20-24).

Abstract, line 15. "In winter, we find channels over 1 m2 in area occurring up to 5 km inland, which shows that the common inference of zero winter freshwater flux is invalid" You could have non-zero flux without channels, so this statement does not follow logically. Change to something like: "We show that the common assumption of zero winter freshwater flux is invalid, and find channels over 1 m2 in area occurring up to 5 km inland."

Wording changed as suggested (p.10, lines 15-16).

p. 6, 15: regarding the assumption that "surface melt travels straight to the bed at the point of production", it is worth noting that this is reasonable on a heavily crevassed glacier.

Words added to make this point (p.15, lines 15-16)

5   p. 11, 14: it would be useful to cite a typical thickness of the sheet, and a threshold value when the sheet begins to transition to small channels.

Words added to provide some information on typical sheet thicknesses (p.21, line 13; p.22, line 1). We have not, however, quoted a value for when the transition to channels happens as this transition is not simply dependent on the sheet thickness and occurs at different values in different parts of the glacier.

Section 3.2: Did the modelled drainage system reach steady state by the end of the 3 month simulations? Additionally, more detail is needed in the caption to Fig. 4: Panels b & c: are these pictures of the end of the simulation? Panels d & e: what days of the simulation are shown? Are these for 'maximum' conditions?

We have clarified that the simulations did not reach a steady state (p.24, line 4) and expanded the caption for Figure 4 (p.23,

15  line 3) to state more clearly what the panels are showing.

Section 3.3 and Table 3: See comments above on presenting results from integrated Daily runs. How do the overall mean values of the Daily runs compare with the Average runs? Can simulations based on seasonal averages yield good approximations of seasonal average outputs (e.g. location of plumes and melt-undercutting totals), or does system non-linearity

20  mean that daily simulations are necessary? Add data to Table 3 and present results in Section 3.3, plus appropriate discussion in Section 4.2 & 4.3.

We have added a row to Table 3 (p. 21) detailing integrated plume melt across all simulations and included text in Sections 3.3 (p.29, lines 13-14) and 4.3 (p.36, lines 16-21) presenting and discussing this.

25  p. 18, 18: Here you compare the model output with the results of Chauché (2016). Since this source is an unpublished PhD thesis, you need to provide more context here. What methods were used by Chauché? What were the associated errors? Are the current results more or less reliable than those of Chauché?

Text added (p.31, lines 18-20; p.30, line 1) to provide more context on Chauché (2016).

30  p. 19, 4: The deep fjord water is not 'subtropical'. Use 'warm Atlantic Water' instead.

Wording changed (p.32, line 7) as suggested.

Reviewer 2

The main weakness is that the scientific motivation for the study is never clearly presented or addressed.

By extending our comparison to observations throughout the paper, we believe we have addressed this point by making the paper a more useful, constrained modelling study of Store Glacier. We also highlight the text in Section 1, which clearly positions the paper within the relevant theoretical context. Tidewater glaciers are very important in controlling mass loss at both Poles (responsible for 40% in Greenland, the ice sheet considered here), yet we know very little about their subglacial

5   hydrology, owing, on the one hand, to the practical difficulties imposed by their speed, size and the discharge of subglacial meltwater at depth in the fjord making field studies challenging, and, on the other, the complexity of the interlinked processes rendering modelling complex. However, as long-established in glaciology, the structure and behaviour of the subglacial hydrological system is a key control on the motion of the overlying glacier; at tidewater glaciers, through the formation of buoyant plumes, it is also a major determinant of submarine melting at the calving front. Consequently, to address these issues,

10   this paper aims to show that (1) modelling the hydrology and plumes in a coupled model, as this paper is the first to do, is possible, and (2) that this produces some interesting and sensible scientific results that can be of use in their own right. For example, we provide useful indications of the extent of channelisation and plume melting at Store under different degrees of melting and in different seasons, including winter, and, as far as possible, show how well these relate to observations, thus addressing both our aims and contributing to resolving the issues motivating this study.

Page 1, line 16: Here and in a number of other places the authors comment on the "common assumption" that sub-glacial discharge at a tidewater glacier terminus falls to zero in winter. I was not aware of that being a common assumption. Indeed plume models tend to require a non-zero outflow as an initial condition, so all the studies I know of that are based on plume theory have a background flow in winter by default. Perhaps the authors could provide specific citations to the studies that

20   assume zero outflow?
We have added a couple of references at the first mention of the assumption in the main body of the article (p. 31, line 14).

Page 6, line 5: I'm not sure I follow these equations. Are they specific to the calving front? In that case shouldn't the density in (3) be the seawater density?

25   Yes, these equations are specific to the boundary condition at the calving front, be this in a lake or a fjord. In this case, therefore, the relevant density would be of seawater. We have added text (p.15, line 7) to clarify this.

Table 1: How were these parameter values chosen? How sensitive are your results to those choices?
The choosing of the parameter values is explained in the text preceding Table 1 (p.15, lines 18-23). The values were taken

30   from previously published work (Gagliardini and Werder, 2018) and sensitivity analysis of the GlaDS model was undertaken by Werder et al. (2013). We have added text (p.15, lines 20-23) to the paper to clarify this.

Figure 2: I think this figure would be more informative if one panel showed temperature in winter and summer (on the same scale), while the other showed salinity in winter and summer.

Changed as suggested (p. 19).

Page 12, lines 1-4: I don't agree with this statement. If the bed topography controlled the hydraulic potential gradient, the water would pool in the deepest part of the bed.

5  Agreed, thank you for suggesting this clarification. We meant that the hydraulic potential gradient is mainly controlled by the bed topography, though not completely. The ice thickness is, of course, also a factor and we have clarified this point in the text (p. 22, line 6).

Results: The text in this section could be reduced significantly. Much of it presents information that is readily available in
10  Table 3. The reader should simply be told why those numbers are of interest. A detailed quantification of their relative sizes is unnecessary.

Thank you for this suggestion. We have considered ways in which this section could be reduced in length and have removed what were clearly some unnecessary numbers on page 24 (lines 5-10). It is, however, not clear to us which specific parts of the remaining text the reviewer would like us to reduce or remove, nor what exactly is too long and what is currently appropriate
15  in length. We think the text is appropriate, given the journal guidelines, and so have otherwise chosen to retain it in its original form. The text is there to make clear the magnitude of the changes between winter and summer and between the different summers, as well as serving to reduce the amount of time the reader has to spend referring back a few pages to Table 3. We would be happy to consider further changes if some more specific recommendations were made.

20  Figure 4: You never say whether the "Daily" results are from the end of the summer, or the time of peak meltwater input, or perhaps the time of peak meltwater discharge.

See response to earlier referee comment on Section 3.2, above.

Page 18-19: This is one of the few places where any comparison with observation is made. The results do not compare very
25  well: the grounding line flux is at the low end, but the calving front melt rate is an order of magnitude smaller. However, the observational numbers appear to be poorly constrained. When were the observations made? Were there no analogous measurements made in summer? A little more discussion of is called for.

We have added further discussion of the context surrounding the winter observations (p.31, lines 18-20) and some text to Section 4.3 (p. 36, lines 12-15) to deal with the measurements made in summer. These were not analogous, being derived from
30  side-scan sonar rather than CTD and ADCP data fed through a model, but offer a useful additional constraint. We also note (p.32, line 1) that modelled plume melt rates being lower than observed ones is a pervasive problem in the field (see, e.g. Sutherland et al., 2019), so the mismatch here is not surprising.

Page 20, lines 16-19: Another very cursory comparison with observation. Is the nature of the drainage system the only result that can be used to validate the model? It would help to plot the location of the observations in Figure 5.

The location of the observations is now plotted in Figures 5d and 6d (pp.25-27) and the comparison to observations has been expanded (p.33, lines 22-29).

**Page 22, lines 5-6: Aren't these also results that could be compared with observation? Are sediment-laden plumes seen in satellite imagery obtained in the summer months? How do the times and locations of their appearance compare with your model results?

We have added some text comparing the modelled and observed locations of plumes (p.36, lines 22-28).

Page 22, lines 25-27: The sub-linear relationships discussed are between sub-glacial discharge and melting. I am not aware of any study that has related surface melting (except in the average sense) directly to sub-glacial melting for the simple reason that there is a delay before surface melt emerges at the grounding line. Your Figure 10 does not take this into account, so I'm not sure what the point of showing it is. If you related discharge to melt, I assume you would see the same sort of relationship as others, since it would be a product of your plume model. More useful would be if you could show some sort of correlation between surface and submarine melt with some simple processing (maybe smoothing and a lag) applied to the surface melt signal. That would be a step towards a simple parameterisation of the overall impact of the sub-glacial hydrological network.

We agree that the accompanying paragraph to Figure 10 conflates subglacial discharge and surface melt and have rewritten it to remove this confusion (p.36, lines 31-33), but there are several studies (e.g. Stevens et al., 2016; Carroll et al., 2016; Mankoff et al., 2016; Christoffersen et al., 2012; Slater et al., 2019) that do explicitly assume a direct relationship between surface melting and plume melting without any lag. We therefore feel that showing Figure 10 is valuable to make the point that, in this case, a direct, unlagged relationship is not found. Exploring this relationship further, however, is beyond the scope of this paper.

25 Figure 12: You show the differences between the results obtained with a coarse and a fine model grid, but has the solution converged on the finer mesh, or would further refinement give different results again?

This issue is also analysed in Werder et al. (2013), where they demonstrate that the mesh resolution makes a few percent difference at most in results when GlaDS is run over realistic topography; we have added text to make this clear (p.39, lines 13-15). We also tried several different mesh resolutions in the initial work for this paper and a finer mesh resolution than the
30 one eventually chosen both significantly increases model run time and generates numerical instabilities that crash the model, so was not pursued further, being impractical.

Page 25-26: There is a suggestion here that the agreement between modelled and observed plume locations (see ** above) is poor. It would be more honest to actually show this comparison, especially if the discrepancy can be explained. But if explanation is the "unrealistic" grounding line, why not use a more realistic one?

We have expanded our discussion of the contrasts between observed and modelled plume locations in Sect. 4.3 (p.36, lines 22-28) to give more detail on the mismatch; given the relatively simple nature of this, we feel an additional figure is unnecessary. We agree that using a more realistic grounding line would be ideal, but it is not a straightforward change to implement independently of the calving code we are currently working to integrate with the model, hence our decision to omit doing so for this study.

Page 26, lines 13-14: There is a mention of parameter uncertainty in the plume models, but no mention here or anywhere else about parameter uncertainty in the sub-glacial hydrology model.

We have added a reference to uncertainty in the GlaDS parameters and discussed our planned solution of this by conducting a full validation exercise as part of future work (p.40, lines 19-24). We have also recently submitted an abstract to AGU 2019 on exactly this topic.

Page 26, lines 22-24: This claim is really not supported by the paper. There is very little comparison with observation, and the comparisons that are made show significant discrepancies. That is a major issue with the paper.

We have rephrased the sentence in question (p.40, line 29) to tone down this claim a little. But, having expanded the comparisons to observations made throughout the paper, we feel that the spirit of this statement remains valid. The model is by no means perfect, but its current failings are expected based on its simplified state. We acknowledge the lack of a full validation exercise undertaken as part of this study, but re-emphasise that this is something we intend to undertake, as demonstrated by the submission of our AGU abstract, and subsequently publish, with the fully coupled version of the model.

Page 27, lines 3-8: Are these potentially testable results? If you have measurements of water properties in the fjord, can you diagnose the relative inputs of sub-glacial meltwater versus that produced by melting of the calving front?

It is possible that this could be achieved, but it is beyond the scope of this work.

**List of changes (page and line numbers refer to tracked document below):**

- p.10, line 15: Wording changed from "In winter, we find channels over 1 $m^2$ in area occurring up to 5 km inland, which shows that the common inference of zero winter freshwater flux is invalid" to "We show that the common assumption of zero winter freshwater flux is invalid, and find channels over 1 $m^2$ in area occurring up to 5 km inland in winter"
- p.10, line 18: Removed "annual"
- p.10, line 19: Replaced "outputs" with "we"

- p.10, line 19: Removed "in winter"
- p.10, line 24: Added "of water"
- p.10, line 25: Replaced "outflow" with "subglacial discharge"
- p.15, line 7: Added "at the calving front (i.e. of seawater in this case)"
- p.15, line 15: Added ", which is reasonable on a heavily crevassed glacier such as Store"
- p.15, line 20: Added "and a sensitivity analysis of the model to these parameters"
- p.15, line 22: Added "An additional sensitivity analysis was not undertaken here as being beyond the scope of this study."
- p.19, Fig. 2: Figure altered to have panel a showing salinity in both winter and summer and panel b showing temperature in both winter and summer
- p.20, line 2-4: Caption to Figure 2 rewritten to : "Fig. 1 – Ambient fjord salinity and temperature profiles used as input to the plume model (Chauché 2016). Winter conditions from CTD cast on 02/03/13 approx. 10 km from the calving front; Summer conditions from CTD cast on 02/08/12 approx. 1 km from the calving front. (a) Salinity in winter and summer; (b) Temperature in winter and summer."
- p.21, Table 3: Added row on total plume melt
- p.21, line 13: Added "(typically up to 1 m near the terminus, progressively dropping to below 0.1 m beyond 100 km inland)"
- p.22, line 6: Added "and the ice thickness"
- p.22, line 17: Replaced "grounding line flux" with "subglacial discharge across the grounding line"
- p.23, Fig. 4: Replaced
- p.23, line 3: Rewritten first sentence of caption to Fig. 4 to read "Patterns of typical plume-generated frontal melt across all simulations, showing the 9th August for panels (b)-(e)."
- p.24, line 4: Added ", but does not reach a steady state by the end of either simulation"
- p.24, line 5: Removed "grows from 0.05% in winter (run Winter) to 12.1 % by the end of the SummerAverage12 run, an increase of"
- p.24, line 6: Added "grows by" and "through the summer"
- p.24, line 7: Removed ", from 0.04 m$^2$ to 9.84 m$^2$"
- p.24, line 8: Removed ", from 0.0008 to 5.32 m$^3$ s$^{-1}$"
- p.24, line 10: Removed ", from 0.05% to 6.75%"
- p.25, Fig. 5: Updated
- p.26, line 2: Added "(red dot shows S30 study site from Young et al. (2019))"
- p.27, Fig. 6: Updated
- p.28, line 2: Added "(red dot shows S30 study site from Young et al. (2019))"

- p.29, line 13: Added "The total amount of melt generated by plumes, however, increases slightly in the daily-forced simulations compared to the average-forced ones, by a little under 2% in both 2012 and 2017."

- p.31, line 14: Added "(e.g. Carroll et al., 2015; Slater et al., 2018)"

- p.31, line 18: Added "using CTD and ADCP data gathered in winter 2012-13 as inputs to the Gade (Gade, 1979) and Motyka (Motyka et al., 2003)  models of fjord circulation and melting. It should also be noted that modelled melt rates from plumes consistently underestimate observed melt rates (e.g. Sutherland et al., 2019); this is a pervasive problem in plume modelling, so it is to be expected that we find a similar result."

- p.32, line 1: Removed "but"

- p.32, line 2: Replaced "grounding-line flux" with "subglacial discharge"

- p.32, line 6 : Replaced "grounding-line flux" with "subglacial discharge"

- p.32, line 7: Changed "warm subtropical waters" to "warm Atlantic water"

- p.33, line 22: Added "Young et al. (2019) posited the existence of a channelised drainage system forming up to, but not beyond this point, based on observed velocity patterns from radar and GPS measurements, with a pronounced slowdown occurring at lower elevations on Store in the summer. Doyle et al. (2018), meanwhile, suggested that persistent high pressure and rapid drainage in boreholes at the site were best explained by them tapping in to an extensive distributed drainage system. Our results for summer 2017, a better comparison for observed melt in 2014-15, concur with this pattern, with significant channel growth ceasing around the 30 km mark in the region of the study site, but with a major distributed sheet drainage pathway predicted to lie in its vicinity (red circle on Fig. 5d and 6d)."

- p.36, line 12: Added: "It is also important to note that our mean maximum plume melt rates for all summer simulations (Table 3) accord well with the observed summer melt rate at Store of $3.4\pm0.7$ m d$^{-1}$ from Chauché (2016), measured using side-scan sonar in summer 2012, and with other modelling studies for Greenlandic glaciers (Xu et al., 2013)."

- p.36, line 16: Added "This slight reduction in concentration of melt in the largest plumes in the daily-forced runs also explains the very slight increase in total plume-induced melting (on the order of 2%) found compared to the average-forced runs, as the marginal favouring of the distributed sheet-driven plume spreads higher melt rates over a larger area. However, the difference is very small, and suggests that, if operating glacial hydrological models at longer temporal and/or larger spatial scales, averaged inputs yield similar outputs to daily-resolution data. Whether this remains the case in a fully-coupled simulation would be an interesting target for future work."

- p.36, line 22: Added "A further possibility for validation is provided by the location of the plumes: visible plumes at Store have been observed persistently about 2 km in from the southern margin of the terminus (i.e. about one third in from the right of Fig. 4) and intermittently in the northern embayment (a similar distance in from the left of Fig. 4) (Ryan et al., 2015). Our model predicts the intermittent northern plumes well, but does not produce a persistent plume at the observed location on the southern half of the terminus. Rather, the modelled plumes are more mobile and do not persistently occupy one location. The reasons for this are considered in Sect. 4.5, below."

- p.36, line 31: Added "However, when considering surface melting, many studies assume a direct relationship between this and subglacial discharge, and, consequently, plume melting (e.g. Carroll et al., 2016; Mankoff et al., 2016; Stevens et al., 2016; Slater et al., 2019)."

- p.36, line 34: Added ", though,"

- p.37, line 2: Added "therefore"

- p.38, line 3: Added "Note the correlation between surface melt and water pressure."

- p.39, line 13: Added "This is further supported by the mesh dependency analysis undertaken by Werder et al. (2013) for GlaDS, which shows little variation in results in the presence of realistic topography."

- p.40, line 20-24: Rewritten to "A similar problem applies to the parameters used for GlaDS – observational difficulties mean they are currently poorly constrained, but we hope to improve this by undertaking a full validation exercise, through comparison with an independently derived dataset of calving events at Store, upon the completion of development of a coupled ice-hydrology-plume-calving model, which is the focus of our future work."

- p.40, line 30: Removed: "recreates well"; added "shows promise in realistically recreating"

- p.41, line 23-30: Added "Author Contribution" and "Competing Interests" sections

- p.42, line 2: Added "and PC"

- p.42, line 6: Removed "and Nolwenn Chauché" and "and the ambient fjord data, respectively"

[revised manuscript text omitted]

---

## Referee Report (RR1)

Tidewater glaciers are largely responsible for controlling future global sea level contributions from the Greenland Ice Sheet. However, several critical components of tidewater glacier systems are still poorly understood, which limits our ability to form predictions of future sea level. In particular, we lack a clear understanding of the subglacial hydrologic system beneath Greenlandic tidewater glaciers - particularly within the main trunk and near the terminus - and its time-evolving relationship with surface melt production, submarine melting at the terminus and glacier dynamics. Cook et al. present an exceptional paper that uses modeling solutions to help constrain many of these important and outstanding questions.

The analysis reveals a richly detailed subglacial hydrologic system beneath Store Glacier, including the contributions of both distributed and channelized drainage, and evaluates its sensitivity to surface meltwater and its impact on plume melting and basal water pressure. The authors make a convincing and novel case that the wintertime ]system is under appreciated and that summertime surface meltwater production is weakly correlated with plume melting, but strongly correlated with basal water pressure. In short, these results are critical to our understanding of the largely unseen subglacial system and its impact on tidewater glacier dynamics.

The manuscript and analysis is well written, logical, and clear. The conclusions are well supported by the model and the authors carefully acknowledge model limitations and instances when the model is not well equipped to answer certain questions. The authors could improve their discussion and, perhaps, analysis of the seasonal evolution of the subglacial hydrologic system and plume dynamics at the terminus. I have a couple of suggestions and potential figure ideas that could improve this narrative and strengthen the paper, which are detailed below. I've also included suggestions for several other minor edits. Overall, I would enthusiastically recommend this paper for publication after only minor revisions.

**High level questions and suggestions**

To what degree do plume locations - and corresponding melt rates - vary within a season across the terminus front? The authors comment that the "location of strong convection-driven summer plumes varies as points of discharge from the hydrologic system evolve." To my knowledge, this is an important open ended question, and further analysis would be an interesting addition to the paper. A simple figure (e.g., heat map of the vertical submarine terminus face) showing the evolution of discharge points (i.e., the time and space integration of their location over the summer) would be instructive.

In addition, it would be great to more explicitly connect the vertical panels in figure 4 to map-view features modeled in the distributed and/or channelized system. For instance, why are the largest plumes unassociated with the main channel and sheet flow drainage on the northern and southern glacier margins?

Another important addition would be to show the development of the subglacial hydrologic system (i.e., growth of the aggregated channel area and distributed flow) in the time series shown in figures 7 and 8. This would allow for a more complete view of the degree to which meltwater drives these systems and, in turn, basal water pressure and melting at the terminus. The time evolution of these relationships is an important aspect that perhaps the present model can shed light on.

**Minor revisions**

Page 4, line 10: This is a nice opportunity to motivate why the stable nature of Store is beneficial for the current modeling effort.

Table 3: Please clarify why the average runs do not have "max" time steps.

Page 11, line 10: It would be helpful to the reader if the authors annotated the 3 branches and other relevant features within the subglacial hydrologic system directly in figure 3.

Page 12, line 25: Please specify observations showing flotation.

Figure 4: It would be beneficial in all figures to expand the scale bar to be larger and more legible. It would also help to include a horizontal and vertical scale bar to provide the readers with relevant distances important to the results.

Figure 5 and 6: A scale bar denoting distance - or model domain coordinates - would be helpful here.

Page 17, line 2: "a s" might be a typo?

Great paper - really enjoyed it.

---

## Author Response (AR2)

**Authors' Response to Referee Comments for: Coupled modelling of subglacial hydrology and calving-front melting at Store Glacier, West Greenland**

We would like to thank the third reviewer for their comments on the paper. As a result we have redrawn several figures and included some new ones to improve the clarity of the paper and strengthen the analysis and discussion. We believe this has addressed all the points raised by the reviewer and further improved the paper.

Editorial Comments

Our response (page and line numbers refer to tracked-changes document below)

Reviewer 3

To what degree do plume locations - and corresponding melt rates - vary within a season across the terminus front? The authors comment that the "location of strong convection-driven summer plumes varies as points of discharge from the hydrologic system evolve." To my knowledge, this is an important open ended question, and further analysis would be an interesting addition to the paper. A simple figure (e.g., heat map of the vertical submarine terminus face) showing the evolution of discharge points (i.e., the time and space integration of their location over the summer) would be instructive.

Thank you for this comment. We have added a heat map of plume activity for the daily-forced simulations (Figure 10 in Section 4.3, p. 34), which supports and clarifies our existing discussion of plume locations within this section. We have not performed further analysis of how plume discharge locations shift over the course of a melt season, because, as described in Sections 4.3 and 4.5, we believe the modelled locations and movement of discharge outlets to be partly a product of the lack of a precise representation of the grounding line on the southern, floating part of the terminus. We have, however, included a specific reference in Section 4.5 to wanting to investigate this as part of our future work with a more sophisticated version of the model that includes a more realistic grounding line (p.38, line 16).

In addition, it would be great to more explicitly connect the vertical panels in figure 4 to map-view features modeled in the distributed and/or channelized system. For instance, why are the largest plumes unassociated with the main channel and sheet flow drainage on the northern and southern glacier margins?

We have added some text to Section 4.3 (p. 33, line 5) to discuss this; largely, it is due to the stratification of water in the fjord and the varying depth of the grounding line. Some of it can also be attributed to the dynamism of the hydrological system, as is discussed in Sections 4.3 and 4.5.

Another important addition would be to show the development of the subglacial hydrologic system (i.e., growth of the aggregated channel area and distributed flow) in the time series shown in figures 7 and 8. This would allow for a more complete view of the degree to which meltwater drives these systems and, in turn, basal water pressure and melting at the

terminus. The time evolution of these relationships is an important aspect that perhaps the present model can shed light on.

Thank you for this observation. We have added a line showing median sheet discharge as a proxy for the development of the subglacial hydrological system to figures 7 and 8, and added some discussion of how this informs us as to the state of the subglacial hydrological system in the relevant part of the results and discussion sections (p.24, line 10; p. 26, line 6; p. 30, line 25; p. 31, line 1-5).

Page 4, line 10: This is a nice opportunity to motivate why the stable nature of Store is beneficial for the current modeling effort.

We have added some words about this meaning 'natural' changes at Store do not have to be disentangled from retreat-driven changes to clarify this (p. 8, line 10).

Table 3: Please clarify why the average runs do not have "max" time steps.

We have added some words to clarify this (p. 14, line 10) – for the average runs, the constant nature of the forcing means the end timestep is also the maximum timestep, so we only show one of them.

Page 11, line 10: It would be helpful to the reader if the authors annotated the 3 branches and other relevant features within the subglacial hydrologic system directly in figure 3.

Annotations added.

Page 12, line 25: Please specify observations showing flotation.

Observations now specified (p. 16, line 25).

Figure 4: It would be beneficial in all figures to expand the scale bar to be larger and more legible. It would also help to include a horizontal and vertical scale bar to provide the readers with relevant distances important to the results.

The colour bar has been expanded and added to all panels, and scale bars have been added to panel a.

Figure 5 and 6: A scale bar denoting distance - or model domain coordinates - would be helpful here.

Scale bar added to both figures.

Page 17, line 2: check if "a s" is a typo?

We are unable to see any such typo at the stated location.

**List of changes (page and line numbers refer to tracked document below):**
- p.8, line 10: Added ", as the effects of rapid retreat do not need to be disentangled from 'natural' behaviour"

- p.9, line 14: Added: "where the southern part of the terminus is floating,"
- p.14, line 10: Added "– for the average-forced runs, the end timestep is also the max timestep, so only figures for the end timestep are shown"
- p.16, line 25: Added: "as shown by a marked surface depression behind the calving front denoting the flexion zone"
- p.16, line 26: Deleted: "and"
- p.16, Figure 3: Replaced with new figure with annotations
- p.17, Figure 4: Redrawn with scale bars and bigger colour bars
- p.19, Figure 5: Redrawn with scale bar
- p.21, Figure 6: Redrawn with scale bar
- p.24, line 10: Added "Sheet discharge, as a proxy for the development of the subglacial hydrological system shows a sensitive, slightly lagged response to variations in surface melt in the first half of the model run, but a much more damped response in the second half. The reasons for this will also be discussed in Sect. 4."
- p.25, Figure 7: Redrawn with sheet discharge plotted
- p.25, line 3: Added "red and blue" and "orange and light blue"
- p.25, line 5: Added "Median sheet discharge (dotted line) shows response of subglacial hydrological system to surface melt, and evolution of the system towards greater channelisation over melt season."
- p.26, line 4: Added: "red and blue" and "orange and light blue"
- p.26, line 6: Added: "Unlike in 2012, however, sheet discharge remains sensitive to surface melt variations until around day 70 of the model run, exhibiting a more damped response thereafter."
- p.27, Figure 8: Redrawn with sheet discharge plotted
- p.27, line 5: Added "Median sheet discharge (dotted line) shows response of subglacial hydrological system to surface melt, and evolution of the system towards greater channelisation over melt season."
- p.30, line 25: Added "This interpretation is reinforced by the evolution of sheet discharge in the two summers. In 2012, the strong response to surface melt variations in the first half of the model run shows a predominantly distributed hydrological system with most water transiting through the sheet; the more damped response in the second half shows the formation of a predominantly channelised system where water is preferentially routed through the efficient channels rather than the inefficient sheet. The lagged nature of the sheet's response, however, means it is not possible to see how it responds to the increased melt at the very end of the SummerDaily12 run. In 2017, the pattern of sheet drainage response shows widespread channelisation was not established until towards day 70 (9th August), but was maintained until the end of the model run, as there is little response of sheet drainage to the surface melt fluctuations from day 80 (19th August) onwards."
- p.31, line 1: Added "Table 3"
- p.31, line 3: Added ", with little growth after day 45 (15th July)"

- p.31, line 4: Added ", levelling off from day 63 (2nd August), according with the onset of widespread channelisation shown by the sheet discharge time series (Fig. 7, Fig. 8) as described above"
- p.33, line 5: Added "The summer plume results also reinforce the point made in Sect. 4.1, above, about the importance of the depth of the grounding line for plume activity. There are many areas of strong plume melt towards the centre of the calving front (Fig. 4, Fig. 10), where subglacial discharge is quite low (Fig. 5, Fig. 6, Fig, 9), but the warmer, more saline water at the greater depths (>400 m) reached in this region of the front (Fig. 2) still allow high plume melting to occur without needing much meltwater input. Conversely, despite higher meltwater discharges nearer the margins, the relatively shallow fjord depth and, therefore, colder, fresher ambient conditions (Fig. 2) limit the amount of melting the resulting plumes can achieve. From our model results, consequently, it is clear that the presence and location of warm, saline water in the fjord is equally important for generating plume melt as is sustained subglacial meltwater discharge, in line with buoyant plume theory (Jenkins, 2011; Slater et al., 2016)."
- p.34, line 2: Added ", with several hotspots of plume activity in the southern half of the terminus (Fig. 10)"
- p.34, line 3-5: New figure (Figure 10; plume heat map) added, with caption.
- p.35, Figure 11: Redrawn to have consistent formatting with Figures 7 and 8
- p.36, Figure 12: Redrawn to have consistent formatting with Figures 7 and 8

[revised manuscript text omitted]

---

## Author Response (AR3)

**Authors' Response to Referee Comments for: Coupled modelling of subglacial hydrology and calving-front melting at Store Glacier, West Greenland**

We would like to thank the third reviewer for their comments on the paper. As a result we have redrawn several figures and included some new ones to improve the clarity of the paper and strengthen the analysis and discussion. We believe this has addressed all the points raised by the reviewer and further improved the paper.

Editorial Comments

Our response (page and line numbers refer to tracked-changes document below)

Editor

Table 3: please include the dates when the "max" results were obtained in the caption. It is not explicitly shown in Fig. 7.
The relevant model timesteps have been added to the table caption.

Fig. 3 caption: the red dot appears in panel b and c, not panel a and b. Indeed, I suggest to include the red dot to all panels (bed topo, channel area, and sheet flux) for easier comparisons.
We have redrawn the figure as requested.

Fig. 4: show north and southern sides in the figure or in the caption (it is mentioned at P22L8-10ff).
We have added this information to the caption for Fig. 4.

P18L18: Table 3 shows that water pressure is higher in the winter (2.01 MPa) than in summer (1.13 and 1.3 MPa).
Table 3 shows the effective pressure, not the water pressure, so the text talking about higher water pressures in summer compared to winter is consistent with the data shown in Table 3 (i.e. a higher effective pressure in winter is due to a lower water pressure).

Fig. 5: both end members of channel area is shown with white so It is very hard to see which areas have lowest or highest channel areas. I think the key massage here is high variability rather than location-location variations, so I can accept this color bar, but if possible please change it to a different scheme so that end members are shown in a different color. Probably a colobar with single color (e.g. white to dark blue) works fine here.
Figure redrawn as suggested with a white-to-blue colour bar.

Figs. 5 and 6: Please add the red dot showing S30 study site to panels a-c, which ease the comparison between panels.
Both figures redrawn with red dot on all panels.

Fig. 7/8 caption: clarify "median" as "median over the entire basin" or such. in the text, it is clear enough but many people read figures first.

We have modified the captions to include this information.

**List of changes (page and line numbers refer to tracked document below):**

- p.13, line 10: Added "– this occurred on timestep 60 for SummerDaily12 and timestep 74 for SummerDaily17"
- p.14, Fig. 3: Redrawn
- p.14, Fig. 3 caption: Removed "(b) and (c)"; added "all"
- p.15, line 4: Added "North is to the left, south is to the right."
- p.17, Fig. 5: Redrawn
- p.19, Fig. 6: Redrawn
- p.22, line 17: Added "; taken as the median over the whole model domain"
- p.23, line 10: Added "; taken as the median over the whole model domain"

[revised manuscript text omitted]